# Structurally Human, Semantically Biased: Detecting LLM-Generated References with Embeddings and GNNs

**Melika Mobini**[1]    **Vincent Holst**[1]    **Floriano Tori**[1]    **Andres Algaba**[1]    **Vincent Ginis**[1,2]

[1]Vrije Universiteit Brussel    [2]Harvard University (SEAS)

`{Melika.Mobini,Vincent.Thorge.Holst,Floriano.Tori}@vub.be`
`{Andres.Algaba,Vincent.Ginis}@vub.be`

## Abstract

Large language models are increasingly used to curate bibliographies, raising the question: are their reference lists distinguishable from human ones? We build paired citation graphs, ground truth and GPT-4o-generated (from parametric knowledge), for 10,000 focal papers ($\approx 275$k references) from SciSciNet, and added a field-matched random baseline that preserves out-degree and field distributions while breaking latent structure. We compare (i) structure-only node features (degree/closeness/eigenvector centrality, clustering, edge count) with (ii) 3072-D title/abstract embeddings, using an RF on graph-level aggregates and Graph Neural Networks with node features. Structure alone barely separates GPT from ground truth (RF accuracy $\approx 0.60$) despite cleanly rejecting the random baseline ($\approx 0.89$–$0.92$). By contrast, embeddings sharply increase separability: RF on aggregated embeddings reaches $\approx 0.83$, and GNNs with embedding node features achieve 93% test accuracy on GPT vs. ground truth. We show the robustness of our findings by replicating the pipeline with Claude Sonnet 4.5 and with multiple embedding models (OpenAI and SPECTER), with RF separability for ground truth vs. Claude $\approx 0.77$ and clean rejection of the random baseline. Thus, LLM bibliographies, generated purely from parametric knowledge, closely mimic human citation topology, but leave detectable semantic fingerprints; detection and debiasing should target content signals rather than global graph structure.

## 1 Introduction

Large Language Models (LLMs) (Brown et al., 2020) increasingly synthesize scientific knowledge and autonomously draft literature reviews (Delgado-Chaves et al., 2025; Dennstädt et al., 2024; Gougherty & Clipp, 2024; Kang & Xiong, 2024; Skarlinski et al., 2024; Susnjak et al., 2024). This raises a core question about what they internalize from science: beyond surface form, do they reproduce the structural and semantic regularities of citation behavior that scaffold scholarly knowledge? To situate our evaluation, we connect to broader capability frameworks for LLMs (Chang et al., 2024; McDermott, 1976). Prior work reports that LLM-written bibliographies often look structurally plausible while being semantically unreliable (Tang et al., 2025; Li et al., 2023; Chu et al., 2024; Li et al., 2024a).

We make this tension concrete: How do citation graphs induced by LLM-suggested references compare, structurally and semantically, to ground truth citation graphs? Are LLM-generated graphs realistic enough to pass structural scrutiny yet betray detectable semantic shifts? Here we build on the earlier analyses of LLM bibliographies purely generated from parametric knowledge, i.e., without access to external databases (Algaba et al., 2024; Mobini et al., 2025; Algaba et al., 2025). These studies find close alignment with human citation networks across observable characteristics (title length, team size, citation and reference counts), alongside systematic differences: LLMs reinforce the Matthew effect (higher median cited-by counts, preference for prestigious venues), favor recency, shorter titles, and fewer authors, reduce self-citations, and match focal-paper similarity at the level of off-the-shelf embeddings. Local graph motifs likewise resembled human networks.

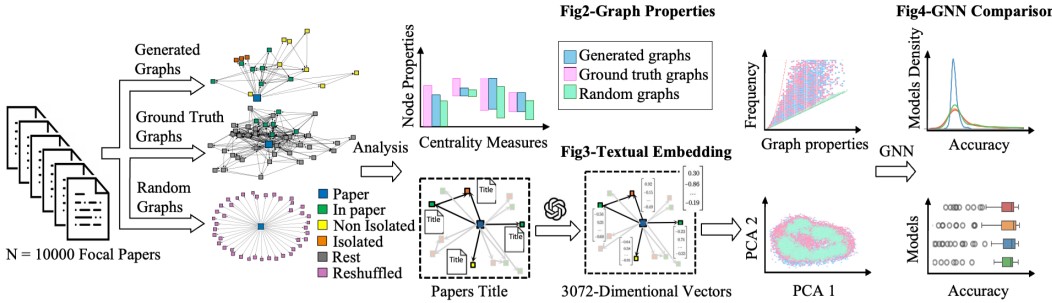

Figure 1: **Overview of the experimental pipeline comparing graph structure and semantic embeddings of LLM-generated and human citation networks.** The study involves comparing the citation networks of focal papers created using ground truth references and LLM-generated references. Random baselines are generated by reshuffling ground truth references. Graph properties and semantic embeddings are compared, and various GNN models are employed to differentiate between generated and ground truth graphs.

What remains unresolved is whether one can reliably distinguish LLM vs. ground truth reference lists using only their induced citation graphs and textual signals.

We address this, see figure 1 with a progressive modeling strategy from interpretable features to deep graph learning. First, we aggregate node-level structural metrics into graph descriptors and evaluate their discriminability with a Random Forest (RF) (Breiman, 2001) over three classes: (i) ground truth graphs built from human references, (ii) generated graphs built from LLM-suggested references, and (iii) domain-matched randomized graphs. These structural features cleanly separate both (i) and (ii) from random baselines but do not separate (i) from (ii) at statistically significant levels. Motivated by this, we next use title-based semantic embeddings with RF, which markedly improves discrimination. Finally, we train Graph Neural Networks (Gori et al., 2005; Scarselli et al., 2005; Zhou et al., 2020; Wu et al., 2021) that learn jointly from structure and node text, yielding further gains. Unlike single-citation audits such as LLM-Check (Sriramanan et al., 2024), we evaluate entire reference lists via their induced citation-graph structure and embedding cohesion.

If LLMs mimic human-like topology yet retain a distinct semantic fingerprint, detection and debiasing should target content signals rather than coarse graph structure. This matters for automated literature-review tools, citation recommendation systems, and "LLM-in-the-loop" scientific workflows.

## 2 RELATED WORK

There is a fast-growing literature on using LLMs as research assistants to support paper search/reading, drafting surveys, and even automating parts of reviewing, which also surfaces recurring concerns about reliability and evaluation in scholarly tasks Lin et al. (2023a); Zhuang et al. (2025); Chen et al. (2025); Wu et al. (2025); Li et al. (2024b); Skarlinski et al. (2024); Susnjak et al. (2024). While most discussion starts from synthetic prose, the same problem appears in more "structured" scholarly artifacts that LLMs increasingly produce, most notably reference lists, where outputs can look superficially plausible while still being systematically wrong, incomplete, or skewed. Recent work has begun to measure how LLM-produced bibliographies compare to human ones, often finding that reference lists can match many coarse bibliometric and graph-structural regularities even when selection differs in more latent ways Algaba et al. (2024); Mobini et al. (2025); Algaba et al. (2025). We position our contribution in this gap by using the induced citation graph and show why topology-only approaches can be weak, motivating an approach that fuses network structure with semantic representations.

Another strand comes from scientometrics and citation network analysis, which treats citation structure as a signal of how knowledge and attention organize at scale, where we ask what kinds of distortions might appear when LLMs mediate scholarly discourse Lepori et al. (2025); Bai et al.

(2025). This connects naturally to work on citation generation and citation recommendation: as LLMs are increasingly asked to "suggest references," they effectively become recommenders whose choices may feed back into discovery, reading, and ultimately credit assignment. Recent studies already point to systematic disparities in who gets recommended and who gets cited—across demographics, geography, and majority/minority status—raising the prospect that LLM-driven reference selection could amplify existing inequalities in visibility Tian et al. (2025); He (2025); Liu et al. (2025).

## 3 CONSTRUCTING CITATION NETWORKS

We use the open-source dataset from Algaba et al. (2025) which consists of $10,000$ focal papers (with $274,951$ accompanying references) sampled from the SciSciNet (Lin et al., 2023b) database. The sample is drawn from papers which are published in Q1 journals between 1999 and 2021, have between 3 and 54 references, and have at least one citation. Additionally, the selected focal paper must have a defined "top field," a valid DOI, and an available abstract. For each focal paper, GPT-4o is prompted to suggest references based on the paper's title, authors, publication year, venue, and abstract. The number of requested references corresponds to the number of ground truth references cited by each focal paper. The existence of the LLM-generated references is determined by cross-verification with the SciSciNet (Lin et al., 2023b) database through fuzzy matching with titles and authors, using conservative similarity thresholds. For more details, we refer to Algaba et al. (2025). In a robustness analysis, we repeat the entire generation pipeline with Claude Sonnet 4.5, using the same prompts, requested reference counts, and fuzzy-matching filters as for GPT-4o. This yields a second set of LLM-generated citation graphs paired to the same focal papers. We reuse the same graph construction and structural descriptors.

As shown in Figure 2 the focal paper forms the primary node (blue node), with references categorized into:

- Green nodes: References cited by the focal paper and suggested by GPT-4o.
- Yellow nodes: Non-isolated GPT-generated references, not cited by the focal paper but connected to other references.
- Orange nodes: Isolated GPT-generated references, neither cited nor connected to other references.
- Grey nodes: ground truth references not suggested by GPT-4o.

Edges between nodes represent citations retrieved from the SciSciNet dataset. Each focal paper initially has a full graph comprising ground truth and GPT-generated references. This full graph is then split into two sub-graphs:

- Generated graph: Contains the focal paper and GPT-generated references (blue, green, yellow, and orange nodes). Some citations produced by the model were not present in the focal paper's ground truth reference list and we represent these exactly as the model predicted, a single edge from the focal paper to each predicted citation node.
- Ground truth graph: Contains the focal paper and ground truth references (blue, green, and grey nodes).

We quantified the semantic role of isolated nodes by comparing cosine-similarity distributions for random references, ground-truth references, non-isolated GPT references, isolated GPT references, and references shared between GPT and the ground-truth bibliography, as shown in Appendix Figure 18. To evaluate GPT-generated reference lists, we need a random baseline that matches the visible statistics the model could exploit (e.g., citation frequencies, publication years, topical labels) while deliberately breaking the latent citation structure of the ground truth graph. We therefore introduce a random baseline on the field level. In line with (Lin et al., 2023b), field labels are derived from the MAG (Microsoft Academic Graph) fields-of-study records: we use the Level-0 fields as "top fields" and the Level-1 fields as "subfields", based on the field assignments provided in SciSciNet. When a paper has multiple field entries, we keep the field with the highest normalized citation count, implicitly assuming that this field is more relevant to that paper. For all focal papers, we then construct a synthetic citation graph by reassigning each focal paper's references

to random papers from the same research field. Concretely, within each (top) field of study, we uniformly permute the ground truth references, i.e., a without-replacement shuffle at the field level, and then attach the permuted references back to their original focal papers. This preserves each focal paper's out-degree (reference count) and the field-level distributions of citation frequencies and publication years, while potentially destroying the latent citation structure that makes ground truth graphs informative. We also construct an analogous subfield-level random baseline using the same procedure, but now within SciSciNet's finer classification into 292 subfields (nested within 19 top fields in total). This analysis yields qualitatively similar results, see Appendix figures 12 and 13. Finally, we construct a temporally constrained random baseline in which resampled references are drawn from the same field but restricted to publication years $\leq$ the focal paper's year. In the original field-level reshuffling, temporal order (reference published before the focal paper) is preserved for roughly 80% of focal–reference pairs, whereas under the temporally constrained baseline fewer than 1% of edges violate temporal order, compared to about 6% of GPT-generated suggestions that point to papers published after the focal paper. Structural scatter plots and RF performance for this temporal baseline (see Appendix Figure 14 and table 11) closely mirror the original random baseline: random graphs remain clearly separable from both ground truth and LLM-generated graphs, even under explicit time constraints.

To ensure a fair comparison, we randomly remove a subset of references from ground truth graphs and random graphs to match the size of the generated graph. We remove 779 graphs as we only kept graphs with at least one existing generated reference from GPT-4o graphs and 89 from Claude graphs, and end up with 9,218 and 9908 graphs per group for GPT-4o and Claude respectively. We replace each directed edge with an undirected, yielding a simple graph. Doing so, the comparisons reflect the topological organization of the network (who is connected to whom and patterns of clustering/centrality), rather than directionality artifacts or trivial in/out-degree differences.

## 4 CLASSIFYING CITATION GRAPHS USING GRAPH TOPOLOGY

We conduct a comprehensive quantitative analysis of each citation graph to characterize global topology and test whether structure alone separates ground truth from GPT-generated instances. We deliberately restricted ourselves to interpretable descriptors that scale well and transfer across graphs of different sizes: degree centrality, closeness centrality, eigenvector centrality (Freeman, 1978), clustering coefficient, and edge count. These capture complementary local and global aspects of citation structure (connectivity concentration, path accessibility, prestige amplification, local triadic closure, and overall density). For node-level quantities we used normalized definitions to ensure comparability when graph order and density vary; graph-level descriptors were formed by aggregating node metrics using summary statistics (mean, median, interquartile range, and maxima-to-mean ratios), so that scale differences do not require ad hoc rescaling.

Figure 2 reports distributions over all 9,218 graphs. The box plots reveal that centrality profiles in GPT-generated graphs closely match ground truth in both medians and IQRs, with similarly occasional high-degree nodes. In other words, the concentration of visibility and reach—how much "attention mass" accumulates on a few nodes—looks nearly identical across the two sets. For clustering, medians are again comparable, but the ground truth set exhibits a wider IQR indicating that the model sometimes produces unusually dense local closure. By contrast, the field-matched random graphs remain tightly centered at low clustering, with narrow dispersion, underscoring a lack of triadic closure.

Further, Figure 2(b) shows a joint view of average degree centrality and clustering coefficient. Ground truth and GPT-generated graphs overlap almost entirely within structural constraints: as average degree rises, the combinatorics of shared neighbors make triadic closure more likely, pushing points upward in clustering; simultaneously, extremely high average degree with only moderate clustering is scarce, which explains the low number of points in that portion of the plane. Random graphs, in contrast, collapse toward the low-degree/low-clustering corner, reflecting their tree-like sparsity.

Additional pairwise relations reinforce this picture. The coupling between mean degree and the maximum-to-mean degree ratio in Figure 2(c) is similar in ground truth and GPT-generated graphs, indicating a comparable extent of hub dominance relative to overall connectivity. Likewise, edge count vs. mean degree and edge count vs. node count trace families of trends consistent with realistic

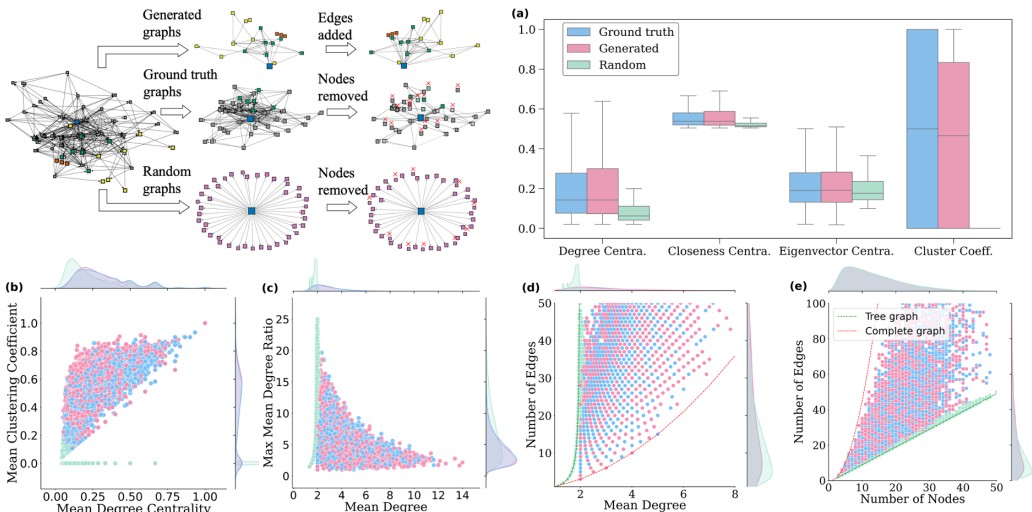

Figure 2: **Pipeline to generate the ground truth and GPT-generated citation graphs and feature histograms** (a) The distributions of four node-level metrics computed for each graph. The ground truth and generated references for degree centrality indicate similar medians and interquartile ranges, suggesting the frequent presence of high-degree hub nodes. Random graphs distribution of degree centrality values differs in shape, as their hub nodes are only the focal papers. For closeness centrality ground truth graphs and generated graphs curves cluster in the upper half of the scale, indicating consistently short geodesic distances. Random graphs show lower values with minimal variance, revealing their sparse and less interconnected structure. All three sets display a positive skew for eigenvector centrality with slightly lower median for random graphs. The clustering coefficients of ground truth and GPT graphs have similar mid-range medians, while random graphs are centered around zero, confirming the absence of triangle-based local cohesion. (b) The scatter plot with marginal histograms, each point represents one graph, ground truth and generated graphs both concentrate in the mid-range of degree centrality and clustering. Random graphs, by contrast, cluster tightly at low values on both axes, showing their consistently sparse and unstructured topology. Marginal density plots along the top and right panels show the univariate distributions of degree centrality and clustering coefficient, respectively. (c) The mean node degree in each graph against its max-to-mean degree ratio in random graph types concentrate at low mean degrees, with ground truth and generated graphs showing nearly identical peaks in both histogram and density. In ground truth and generated graphs, mean degree and ratio trade off smoothly whereas mean degree increases, the relative prominence typically diminishes. Random graphs differ by showing a wider range of ratio values and a consistently lower mean degree. (d) The total number of edges is plotted against its mean node degree. The red dashed line indicates a complete graph, while the green dotted curve represents a tree graph. Both the generated and ground truth graphs show a clear upward trend, graphs with more edges unsurprisingly have higher average degrees, and their point clouds largely overlap. This shows that the generative model accurately reproduces the empirical scaling between graph density and per-node connectivity, mimicking the scaling of ground truth graphs, which lies between sparse tree graphs and fully-connected complete graphs. Random graphs break this pattern by clustering tightly at low average degrees, indicating a sparse attachment model that lacks the varied connectivity patterns seen in other sets. (e) The scatter of Edges vs. Nodes both ground truth and generated graphs lie predominantly between these extremes by covering the entire range, whereas the random graphs resemble tree-like growth representing uniformly minimal connectivity.

densities: both human and GPT-generated graphs fall between tree and complete-graph bounds, whereas random graphs tend to cluster near the tree limit. The near-complete overlap of ground truth and GPT-generated point clouds across these projections indicates that the generative process reproduces the characteristic coupling between global connectivity and local cohesiveness that we observe in human bibliographies, not merely matching marginal distributions but also multivariate relationships among descriptors.

| Graph properties | Ground truth vs. GPT | Ground truth vs. Random | GPT vs. Random |
|---|---|---|---|
| Mean Accuracy | $0.6079 \pm 0.0058$ | $0.8956 \pm 0.0024$ | $0.9275 \pm 0.0059$ |
| Mean F1-score | $0.6061 \pm 0.0055$ | $0.8946 \pm 0.0026$ | $0.9272 \pm 0.0060$ |

Table 1: **Performance of the RF on graph properties.** The table shows the mean accuracy and F1-score obtained from applying a RF classifier across 10 independent runs with different random seeds, utilizing graph-based features. The dataset was partitioned into training/validation and testing subsets, with a train set of 0.7 and a validation/test size of 0.15. The number of estimators sets to 100. Given the balanced nature of the dataset and the binary classification setting, accuracy is robust, but we also used F1 to minimize false positives and negatives.

Importantly, these similarities persist despite topical matching of the random baseline: even when drawn from the same broad field, random graphs systematically exhibit lower connectivity, minimal clustering, and sparser, star- or tree-like topologies. This separation highlights a qualitative distinction: random graphs fail to instantiate the hub-rich, locally cohesive patterns that empirical citation networks display, whereas GPT-generated graphs do capture these signatures at both local and global scales. In that sense, the structural realism of GPT-generated bibliographies extends beyond single-feature alignment to joint, topology-level constraints.

We then assess whether these structural descriptors suffice for automated discrimination by training a Random Forest (RF) (Breiman, 2001) on the aggregated graph-level features and evaluated three binary tasks: Ground truth vs. GPT-generated, Ground truth vs. Random, and GPT-generated vs. Random (Table 1). Against random baselines, structure is highly informative: we obtain accuracies of $0.8956 \pm 0.0024$ (Ground truth vs. Random) and $0.9275 \pm 0.0059$ (GPT vs. Random), with comparable F1-scores, confirming that realistic citation topology is far from random even under field controls. However, when directly contrasting GPT-generated with ground truth, performance drops to near-chance: Mean Accuracy $0.6079 \pm 0.0058$, Mean F1-score $0.6061 \pm 0.0055$. Within this descriptor family, then, structural properties alone do not reliably differentiate LLM from human reference lists at scale. Running the same experiment with Claude-generated graphs shows the same pattern: near-chance Claude vs ground truth on structure, but again a clean separation from the random baseline see Appendix Figure 5 and table4. Taken together, these results support two conclusions that guide the rest of the paper. First, GPT-generated bibliographies are structurally realistic: their global topology, centrality concentration, and local closure closely track ground truth citation graphs, and they occupy essentially the same region of the structural feature space. Second, any systematic differences that remain are unlikely to be captured by coarse structural summaries; to detect residual signatures, one must look beyond topology—toward textual/semantic signals and models that can fuse node content with structure.

## 5 Classifying citation graphs using LLM embeddings

To complement our structural analysis, we measure content-level alignment by extracting embeddings of the title and abstract for all the focal papers, and embeddings of the titles of the ground truth and GPT-generated references using the OpenAI text-embedding-3-large model. This results in a 3072-dimensional embedding vector for each focal paper and reference which is used as the node features. We then proceed by computing the sum over all ground truth references embedding vectors, over all GPT-generated references embedding vectors, and over all random references embedding vectors for each focal paper. To assess the robustness of our analysis, we do the same analysis with the SPECTER2 (Singh et al., 2023) embedding model which results in a 768-dimensional embedding vector, see Appendix Figure 8 and Figure 10. For titles only, we use the adhoc-query fine-tuned version, and for title and abstract combinations of the focal papers, we use the proximity fine-tuned version.

Figure 3 shows a 2D PCA of the graph-level embeddings with contours showing substantial overlap between ground truth and LLM graphs, while random graphs occupy a distinct region. Importantly, this 2-D projection is purely illustrative: the first two principal components explain only about 6% of the variance in the 3072-dimensional title embeddings, and several hundred components are needed to explain more than 90% of the variance. The classifiers we report operate in this higher-dimensional semantic subspace, not in the compressed 2-D view, see Appendix Figure 16

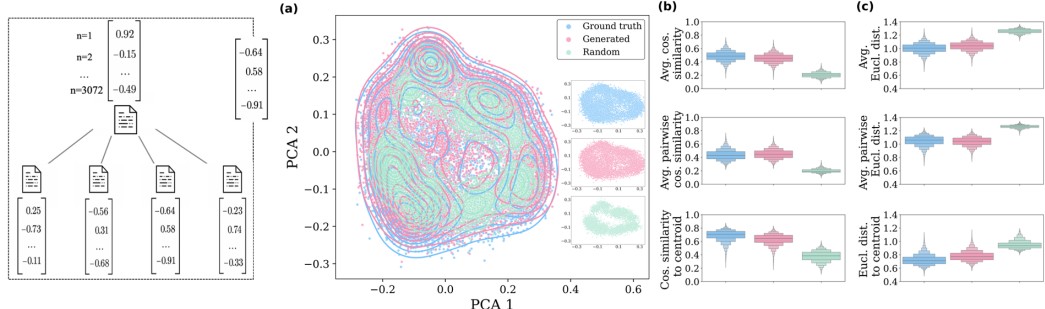

Figure 3: **PCA of graph embeddings with Cosine/EU Distances** (a) Summing 3072-d node embeddings to graph level, visualize the embedding space with 2D PCA (contours). (b) Cosine alignment (node level). Three graph-wise alignment diagnostics: mean of focal with reference, mean of reference with reference, and focal vs. sum of references, capturing how well references align with the focal paper and with each other. (c) Euclidean dispersion (node level). Distance-space counterparts of mean focal with reference, mean reference to reference, and focal vs. sum of references, quantifying semantic spread.

and Figure 17. For every graph we measure the similarities and the distances between nodes. The cosine similarity and distance scores exhibit a strong semantic alignment of generated references with ground truth references compare to weak alignment in random graphs.

We then apply the same RF classifier on the aggregated embedding vectors. As shown in Table 2, classification performance significantly surpasses the one achieved with structural features alone: distinguishing ground truth from GPT-generated graphs reaches a mean accuracy of $0.8346 \pm 0.0063$, compared to $\approx 0.6$ with purely structural metrics. Both ground truth and GPT-generated graphs are well-separated from random baselines, achieving over $0.90$ accuracy in both cases. These effects persist across encoders: GPT-4o references embedded with SPECTER (Singh et al., 2023) and Claude Sonnet 4.5 references embedded with OpenAI/SPECTER remain separable from ground truth references (embedded with SPECTER, and OpenAI/SPECTER, respectively), mirroring the backbone results of GPT-4o references embedded with OpenAI, see Appendix Figure 6 and table 7 and 5 and 8.

In our RF across all trees, the average leaf depth clusters around $\approx 10$ levels—showing that early splits do most of the discriminative work (See Appendix Figure 20). These results suggest that semantic embeddings capture highly discriminative information.

In both the structural and the semantic settings, performance deviated from a field-matched random baseline that simulates bibliographies assembled by uniformly sampling references from the focal paper's area. This pattern provides initial evidence that LLMs retrieve topically relevant references across domains using their parametric knowledge. These observations motivate our next step to evaluate a range of GNN architectures that can jointly exploit topology and semantics to test whether learned message passing (Gilmer et al., 2017) can match or surpass the signal exposed by these simpler baselines.

## 6 Classifying citation graphs using GNNs

**Models** Graph Neural Networks (GNNs) have become the leading deep learning methods to operate on graph-structured data. Standard GNN architectures include Graph Convolutional Networks (GCNs) (Kipf & Welling, 2016), Graph Attention Networks (GATs) (Veličković et al., 2017), Graph-SAGE (Hamilton et al., 2017), and Graph Isomorphism Networks (GINs) (Xu et al., 2018). Each model offers unique strengths in capturing structural information, scalability, and representational power among various applications. Until now, evaluation of these models has primarily focused on traditional citation networks (Shchur et al., 2018). Due to their conceptual diversity and robust performance across various tasks, these models serve as standard benchmarks for a fair and thorough evaluation of GNNs (Zhou et al., 2020; Errica et al., 2019).

| Title embeddings | Ground truth vs. GPT | Ground truth vs. Random | GPT vs. Random |
|---|---|---|---|
| Accuracy | $0.8346 \pm 0.0063$ | $0.9077 \pm 0.0062$ | $0.9527 \pm 0.0017$ |
| Mean F1-score | $0.8345 \pm 0.0063$ | $0.9070 \pm 0.0063$ | $0.9526 \pm 0.0017$ |

Table 2: **Performance of the RF on embeddings.** The table shows the mean accuracy and F1-score obtained from applying a RF classifier across 10 independent runs with different random seeds, utilizing textual embeddings over graphs.

**Experimental setup**   Reliable comparison of GNN models is often hindered by inconsistent experimental conditions, incomplete reporting, and the use of different data splits. Our results are produced using a consistent evaluation strategy to compare different GNN architectures on our dataset. Earlier work has demonstrated that performance in node classification tasks is highly sensitive to the choice of training, validation, and test splits (Shchur et al., 2018). To reduce the influence of such variability, we adopt multiple random seeds and average results across different data splits and initializations. Finally, we report the final results on the validation set for each hyperparameter setup. We present the hyperparameter search grid in the Appendix table 12. For each model we then select the best performing setup and compute the performance on the test set. This offers, combined with the distribution of the validation performance, a transparent way to compare different models over the entire hyperparameter grid while still maintaining a direct comparison on the test set.

Structural attributes are usually incorporated into node features in GNNs to improve performance. This strategy benefits from the expressive power of aggregation functions like summation, but it can also limit a model's ability to adapt to graphs exhibiting unfamiliar degree distributions. For instance, (Cui et al., 2022) found that sum-based aggregation remarkably outperforms mean aggregation in structural node classification tasks, because it maintains neighborhood counts and thus embodies critical structural information. Meanwhile, (Lee & Yoon, 2022) indicate that average node degree is a key feature influencing GNN generalization to unseen graphs, and support for modular update mechanisms to better adapt to graphs with heterogeneous degree distributions. To preserve consistency, all models in our study are trained and tested using the same node feature representations. We performed three sets of pairwise comparisons between graph types: GPT vs. Ground truth, GPT vs. Random, and Ground truth vs. Random once with graph properties and once with semantic based vectors. Following the construction of citation graphs, we extracted structural features and embedding based vectors for each node and assigned these as input features for downstream learning tasks. Each node was assigned a five-dimensional feature vector consisting of degree centrality, closeness centrality, eigenvector centrality, clustering coefficient, and the graph's total number of edges, which is a graph level features but here assigned as node feature in GNN training using the graph for structural features. And a 3072-dimensional feature vector was made for each node in the graphs for GNN embedding-based comparisons. For each pairwise comparison we used a binary label to differentiate between GPT, ground truth or random graphs.

The dataset was split into training, validation, and test sets in proportions of 70, 15, and 15, respectively using stratified splits in such a manner that if a ground truth focal paper appeared in the train dataset, its respective random graph also appeared in the same split set. To ensure reproducibility and consistency across multiple models and experiments, the random seed was fixed during splitting. Although the splits were deterministic, internal ordering was shuffled before training to avoid learning biases tied to sequence. Each model was trained using a consistent optimization protocol. For all learning we used the Adam optimizer (Kingma & Ba, 2014).

**Results**   Figure 4 summarizes the performance of four GNN architectures across our classification tasks, GCN, GAT, GIN, and GraphSAGE. The reported accuracies are the final validation accuracies over the sweep of hyperparameters. We choose to report the full performance on hyperparameters to provide a transparent performance overview instead of simply highlighting the top performer. We do provide the top performing result for completness in Table 3 for the test dataset and the best hyperparameters in Table 13 and 14. The upper row of the figure shows the performance when we only use the graph properties as feature vectors, while the bottom row highlights the performances when feature vectors are the embedding vectors used previously. In this setting, GNNs achieve high separability for GPT vs. Random with a high density across GNN architectures, reflecting substantial structural differences between generated and randomized citation networks. However,

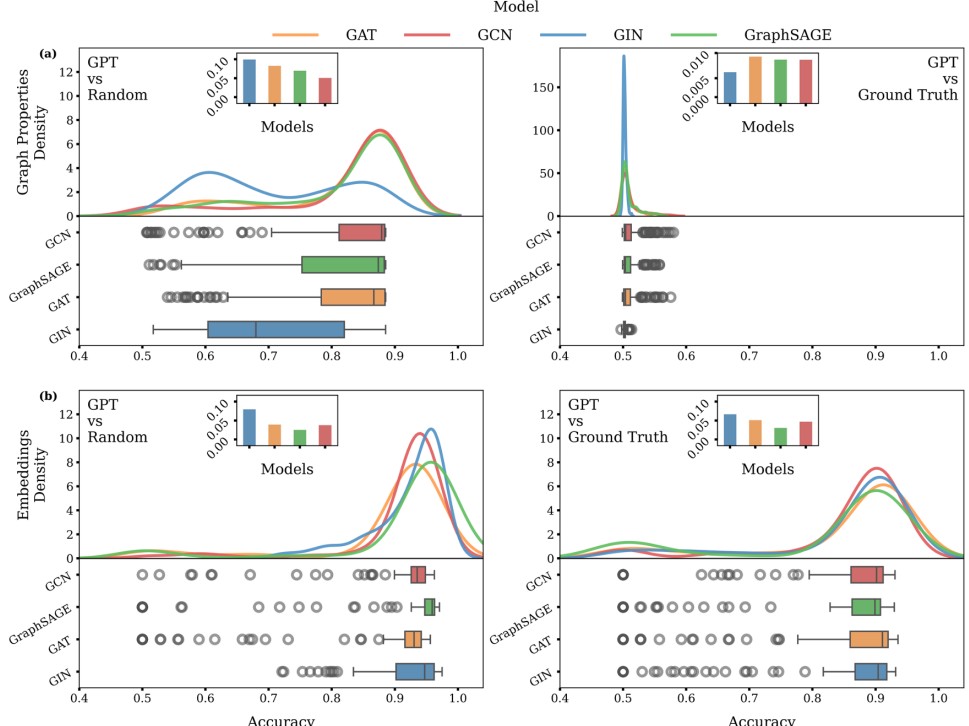

Figure 4: **Distribution of final validation accuracy over the sweep of hyperparameters using different models** Each panel summarizing the performance of four GNN architectures on validation set across our binary classification tasks using different seeds. Above figures show the GNN using the graph properties and the bottom figures are related to the GNN results using embedding vectors. Within every panel, the top axis shows kernel-density estimates (KDEs) of per-run final accuracies, and the bottom axis shows each GNN architecture boxplots medians, interquartile ranges, whiskers, and outliers. The bar plots illustrating the mean standard deviation of accuracy on validation dataset across different models showing consistency over models. Each sweep consists of 500 hyperparameter setups.

performance drops sharply in GNN experiments using graph properties for the GPT vs. ground truth classification, with accuracies clustering around chance level. Using different models could not increase the accuracies, indicating that structural features along with message passing are insufficient for distinguishing LLM-generated graphs from ground truth citation graphs. We also show the results for the GNN performance of the Claude references in the Appendix Figures 7 and 9 and the performance of GPT-4o references using OpenAI embeddings (See Appendix Figure 11, and the performance of GNN in the test set 6. Training on GPT-4o and testing on Claude Sonnet 4.5 yields substantial above-chance generalization for all GNNs (See results in Appendix 8); an RF trained on GPT-4o also reaches $\approx 0.72$ when the generator is swapped at test time (See results in Appendix 9).

When replacing node embeddings with i.i.d. vectors of matched dimensionality, RF/GNN accuracy collapses to chance, showing that the gains are the result of semantic structure rather than the number of features, see Appendix results 15.

To assess robustness, we quantify distributional saturation as runs accumulate using permutation-averaged Wasserstein distance (see Appendix figure 19), showing that distributions saturate early and additional sweeps have marginal impact.

By using embedding based features, we observe an improvement in the GPT vs. ground truth task: all GNN architectures surpass chance performance by a significant margin. This indicates that, although GPT-generated citation graphs mimic the structural properties of ground truth networks, their semantic embeddings encode subtle but learnable differences in language patterns. These results

Table 3: Results on test set ($\nwarrow$ Accuracy, $\searrow$ F1).

| Model | Ground truth vs GPT | | Random vs GPT | |
|---|---|---|---|---|
| | Graph properties | Embeddings | Graph properties | Embeddings |
| GCN | $57.73_{\pm 2.10}$ / $56.83_{\pm 4.10}$ | $93.10_{\pm 0.55}$ / $93.10_{\pm 0.55}$ | $88.51_{\pm 0.31}$ / $88.38_{\pm 0.32}$ | $95.23_{\pm 4.99}$ / $95.18_{\pm 4.16}$ |
| GraphSage | $55.59_{\pm 1.93}$ / $55.27_{\pm 2.18}$ | $93.14_{\pm 0.51}$ / $93.14_{\pm 0.51}$ | $88.51_{\pm 0.27}$ / $88.37_{\pm 0.30}$ | $95.85_{\pm 2.35}$ / $95.84_{\pm 2.38}$ |
| GAT | $57.40_{\pm 2.44}$ / $57.03_{\pm 3.17}$ | $93.78_{\pm 0.43}$ / $93.78_{\pm 0.43}$ | $88.53_{\pm 0.31}$ / $88.41_{\pm 0.32}$ | $95.50_{\pm 0.56}$ / $95.49_{\pm 0.56}$ |
| GIN | $51.71_{\pm 2.70}$ / $47.23_{\pm 6.81}$ | $93.28_{\pm 0.57}$ / $93.28_{\pm 0.57}$ | $88.47_{\pm 0.27}$ / $88.34_{\pm 0.28}$ | $97.39_{\pm 0.23}$ / $97.40_{\pm 0.23}$ |

suggest that the primary signature distinguishing LLM-generated from human-generated citation graphs lies not in topology but in semantic content.

## 7 Discussion

We present a large-scale, paired evaluation of LLM-produced bibliographies against human ground truth, supplemented by a field-matched random baselines. Our analysis proceeds from interpretable graph-level descriptors to content-aware graph neural networks, cleanly decomposing what is captured by citation topology versus node semantics. Across transparent descriptors and structure-only models, LLM-generated citation graphs are essentially indistinguishable from human ones, while the random baseline is cleanly rejected. Incorporating title/abstract embeddings changes the picture: content-aware models reliably separate LLM graphs from ground truth, indicating that residual differences reside in semantics rather than topology. We strengthen our analysis by including further robustness checks. We checked that our results do not depend on the embedding backbone or dimensionality of the embedding vectors, are robust under two additional random baselines controlling for subfield and publication year of the focal paper, and even generalize to cross-generator experiments.

Practically, this hints that auditing and debiasing LLM-generated bibliographies should prioritize content signals (e.g., embedding distributions, topical drift, recency tilt) over coarse structural features. Detection pipelines built on text embeddings or text+graph hybrids are therefore the right tool; structure-only diagnostics will under-detect.

Methodologically, our protocol (paired graphs, domain-matched randomization, and a stepwise path from transparent features to content-aware GNNs) constitutes a general recipe for stress-testing bibliographic authenticity without relying on manual curation.

## 8 Conclusion, limitations and future work

Our work analyses two LLM families over multiple embedding backbones while focusing on title/abstract text rather than full-text. In the current analysis, we focus solely on the parametrically retrieved references, allowing for a stricter lab setting and probing directly the biases of the models. We do not consider the references that would be obtained when models have access to external databases. Future work could probe which semantic dimensions drive separability and what they could mean (recency, prestige, method vs. theory, author overlap). Taken together, our results suggest a simple operating principle: today's LLMs can convincingly mimic the shape of citation, but not yet its semantics and it is therefore in this semantic fingerprint that reliable detection (and eventual correction) must operate.

ACKNOWLEDGEMENTS

This research was supported by funding from the Flemish Government under the "Onderzoekspro-gramma Artificiële Intelligentie (AI) Vlaanderen" program. Andres Algaba acknowledges support from the Francqui Foundation (Belgium) through a Francqui Start-Up Grant and a fellowship from the Research Foundation Flanders (FWO) under Grant No.1286924N. Vincent Ginis acknowledges support from Research Foundation Flanders under Grant No.G032822N and G0K9322N. The resources and services used in this work were provided by the VSC (Flemish Supercomputer Center), funded by the Research Foundation - Flanders (FWO) and the Flemish Government.

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

## APPENDIX

## PROMPTING GPT-4O

The GPT-references in the dataset from Algaba et al. (2025) were generated using the following prompt in gpt-4o-2024-08-06:

The system prompt for each focal paper is:

> *Below, we share with you the title, authors, year, venue, and abstract of a scientific paper. Can you provide $\{n\}$ references that would be relevant to this paper?*

The user prompt contains basic bibliographic metadata and the abstract:

```
Title: {PaperTitle}

Authors: {Author_Name}

Year: {Year}

Venue: {Journal_Name}

Abstract: {Abstract}
```

Here $\{n\}$ is set to the ground truth number of cited works for that focal paper. We subsequently parse the model's suggestions for bibliometric data (title, authors, number of authors, venue, publication year) in a postprocessing step using a second prompt (called via *gpt-4o-mini-2024-07-18*):

> *Below, we share with you a list of references. Could you for each reference extract the authors, the number of authors, title, publication year, and publication venue? Please only return the extracted information in a markdown table with the authors, number of authors, title, publication year, and publication venue as columns. Do not return any additional information or formatting.*

## PROMPTING CLAUDE SONNET 4.5

Following the methodology of Algaba et al. (2025), we generate reference suggestions using the following prompt in Claude-sonnet-4-5-20250929: The system prompt for each focal paper is:

> *Below, we share with you the title, authors, year, venue, and abstract of a scientific paper. Can you provide $\{n\}$ references that would be relevant to this paper?*

The user prompt contains basic bibliographic metadata and the abstract:

```
Title: {PaperTitle}

Authors: {Author_Name}

Year: {Year}

Venue: {Journal_Name}

Abstract: {Abstract}
```

Here $\{n\}$ is set to the ground truth number of cited works for that focal paper.

## CITATION NETWORKS

Let $G = (V, E)$ be a simple undirected graph, where $V$ denotes the set of nodes and $E$ denotes the set of edges. Let $|V|$ denote the number of nodes. For a node $v \in V$ with degree $d_v$, the *degree centrality* $C_D(v)$ can be defined as

$$C_D(v) = \frac{d_v}{|V| - 1},$$

reflecting the fraction of possible connections that $v$ actually has. The *distance* between two nodes $u, v \in V$ is denoted by $d(u, v)$ and equals the numbers of edges in the shortest path between the two nodes. The *closeness centrality* of a node $v \in V$ is determined by the reciprocal of the average shortest path from $v$ to all other nodes:

$$C_C(v) = \frac{|V| - 1}{\sum_{u \neq v} d(u, v)}.$$

Let $C_E(v)$ be the *eigenvector centrality* of node $v \in V$ defined as

$$C_E(v) = \frac{1}{\lambda} \sum_{u \in \mathcal{N}(v)} C_E(u),$$

where $\lambda$ is the largest eigenvalue of the adjacency matrix of the graph.

Let $e(v)$ denote the number of edges among its neighbors. The *clustering coefficient* of $v$ is defined as

$$C(v) = \frac{2e(v)}{d_v(d_v - 1)}.$$

## ALTERNATIVE GENERATORS AND ENCODERS

Extended to Claude and decoupled generator/encoder pairs confirm that the semantic fingerprint is not tied to a single generator/encoder pair.

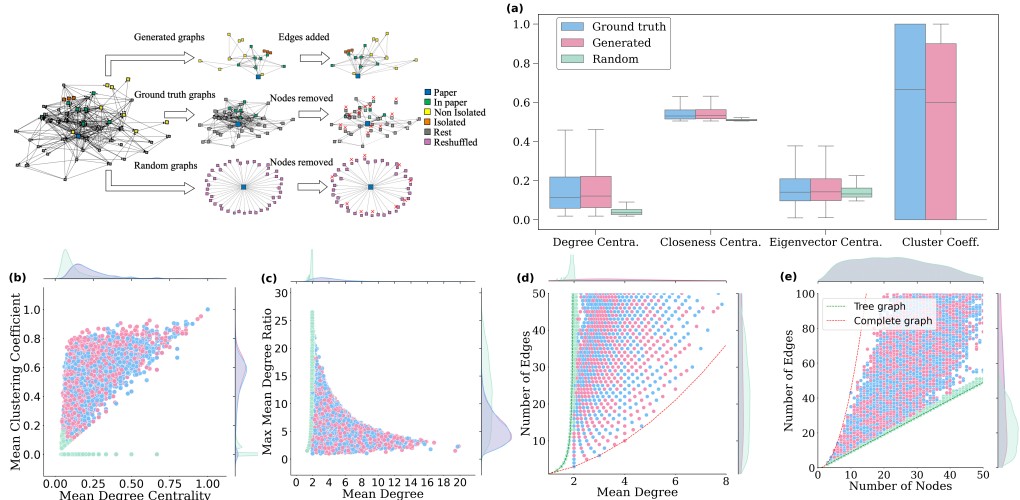

Figure 5: **Structural comparison of citation graphs when references are generated by Claude.** The figure mirrors the pipeline and descriptors used in the GPT-4o analysis. These patterns confirm that Claude bibliographies closely mimic human citation topology and that structural summaries alone cannot reliably expose the generator.

| Graph properties | Ground truth vs. Claude | Ground truth vs. Random | Claude vs. Random |
|---|---|---|---|
| Mean Accuracy | $0.5389 \pm 0.0084$ | $0.9601 \pm 0.0039$ | $0.9701 \pm 0.0024$ |
| Mean F1-score | $0.5386 \pm 0.0085$ | $0.9600 \pm 0.0039$ | $0.9700 \pm 0.0025$ |

Table 4: **Performance of the RF on Claude graphs.** The table reports mean accuracy and mean F1-score over 10 runs with different random seeds, using the same structural descriptors and 70/15/15 train/validation/test split as in the GPT-4o analysis. Structure-only features remain only weakly informative for Ground truth vs. Claude, while ground truth vs. random and Claude vs. random are classified with high accuracy, confirming that Claude reproduces realistic citation topology but is hard to distinguish from human graphs based on coarse structure alone.

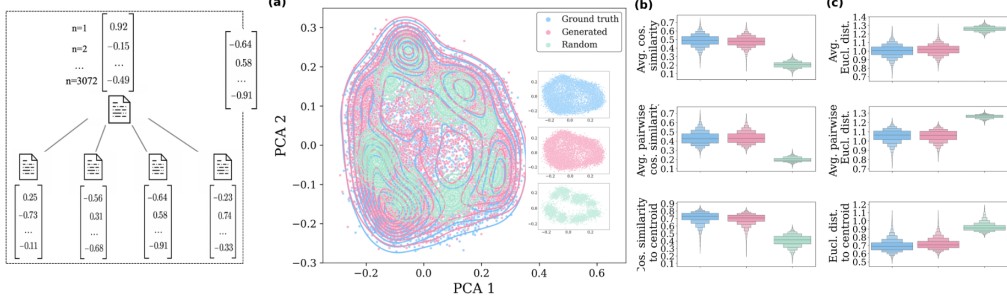

Figure 6: **Embedding-space comparison for Claude graphs using OPENAI** Semantic comparison of citation graphs with Claude-generated references. As in the GPT-4o case, these results demonstrate that Claude retrieves topically appropriate references yet leaves a consistent semantic fingerprint that can be exploited for detection, in contrast to the near-indistinguishability at the purely structural level.

| Title embeddings | Ground truth vs. Claude | Ground truth vs. Random | Claude vs. Random |
|---|---|---|---|
| Accuracy | $0.7688 \pm 0.0067$ | $0.9428 \pm 0.0038$ | $0.9546 \pm 0.0031$ |
| Mean F1-score | $0.7687 \pm 0.0067$ | $0.9426 \pm 0.0038$ | $0.9545 \pm 0.0031$ |

Table 5: **Performance of the RF on Claude graphs using OpenAi embeddings.** Using the same 3072-dimensional OPENAI embeddings and RF protocol as in the GPT-4o experiment, we obtain substantially higher separability than with graph properties: Ground truth vs. Claude reaches $\approx 0.77$ accuracy, while both classes remain cleanly separable from the random baseline. This replicates the pattern that semantic content carries a much stronger detection signal than topology, with slightly lower ground-truth–vs–LLM separability for Claude than for GPT-4o.

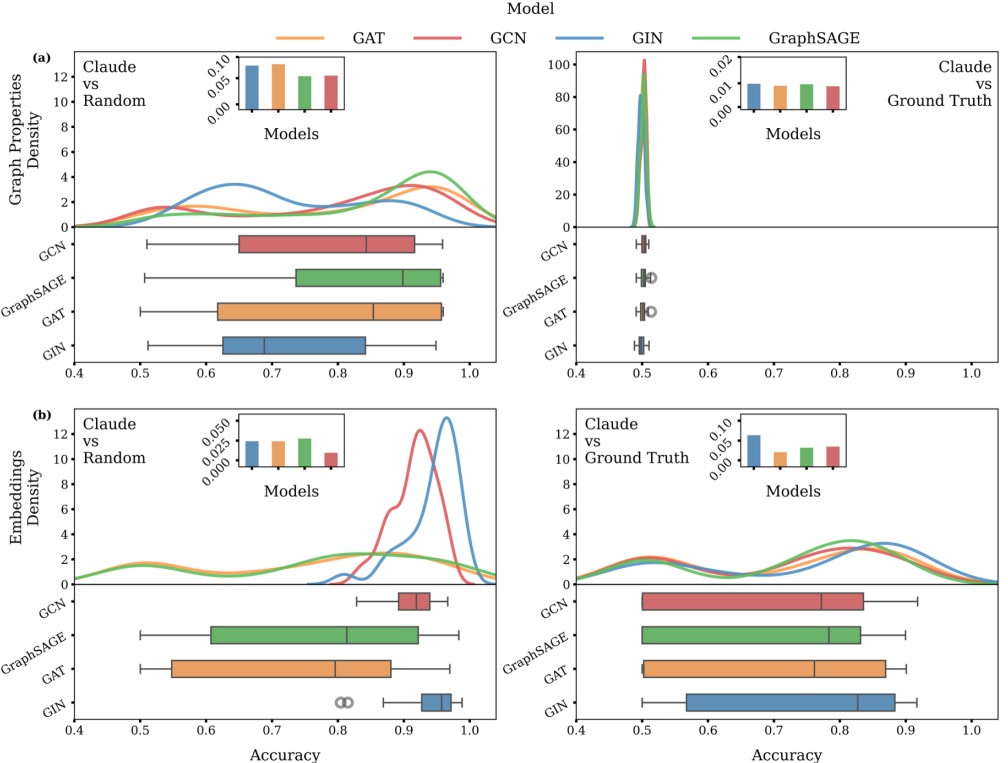

Figure 7: **Performance of the GNN for Claude dataset.** Each panel summarizes the performance of four GNN architectures on the validation set across our binary classification tasks using different seeds. The figures are related to the GNN results using OpenAI embedding vectors. Within every panel, the top axis shows kernel-density estimates (KDEs) of per-run final accuracies, and the bottom axis shows each GNN architecture's boxplots medians, interquartile ranges, whiskers, and outliers.

Table 6: Results on test set for Claude dataset (↖ Accuracy, ↘ F1).

| Model | Ground truth vs Claude | | Random vs Claude | |
|---|---|---|---|---|
| | Graph properties | Embeddings | Graph properties | Embeddings |
| GCN | $50.16_{\pm0.79}$ $47.81_{\pm1.71}$ | $94.50_{\pm0.5}$ $94.50_{\pm0.14}$ | $94.79_{\pm2.00}$ $94.78_{\pm2.03}$ | $95.23_{\pm4.99}$ $96.66_{\pm5.65}$ |
| GraphSage | $50.08_{\pm1.06}$ $43.01_{\pm6.40}$ | $94.34_{\pm0.5}$ $94.4_{\pm0.23}$ | $95.44_{\pm0.37}$ $96.56_{\pm0.45}$ | $96.15_{\pm1.55}$ $96.55_{\pm1.65}$ |
| GAT | $50.25_{\pm0.69}$ $47.47_{\pm4.54}$ | $94.58_{\pm0.55}$ $94.12_{\pm0.34}$ | $95.67_{\pm0.32}$ $96.34_{\pm0.34}$ | $95.50_{\pm0.56}$ $96.44_{\pm0.66}$ |
| GIN | $50.39_{\pm0.93}$ $33.50_{\pm0.41}$ | $94.22_{\pm0.44}$ $94.18_{\pm0.34}$ | $94.81_{\pm1.08}$ $94.79_{\pm1.09}$ | $97.39_{\pm0.23}$ $97.67_{\pm0.44}$ |

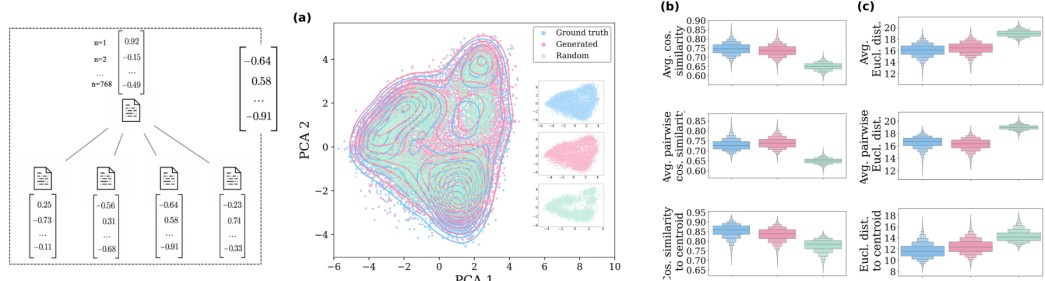

Figure 8: **Embedding-space comparison for Claude graphs using SPECTER.** Summing 768-d node embeddings to graph level, visualize the embedding space with 2D PCA (contours). Semantic comparison of citation graphs with Claude-generated references. As in the GPT-4o case, these results demonstrate that Claude retrieves topically appropriate references yet leaves a consistent semantic fingerprint that can be exploited for detection, in contrast to the near-indistinguishability at the purely structural level.

| Title embeddings | Ground truth vs. Claude | Ground truth vs. Random | Claude vs. Random |
|---|---|---|---|
| Accuracy | $0.6878 \pm 0.0099$ | $0.9289 \pm 0.0058$ | $0.9421 \pm 0.0039$ |
| Mean F1-score | $0.6873 \pm 0.0100$ | $0.9286 \pm 0.0059$ | $0.9419 \pm 0.0039$ |

Table 7: **Performance of the RF on Claude graphs using SPECTER embeddings.** The table shows the mean accuracy and F1-score obtained from applying a RF classifier across 10 independent runs with different random seeds, utilizing textual embeddings over graphs.

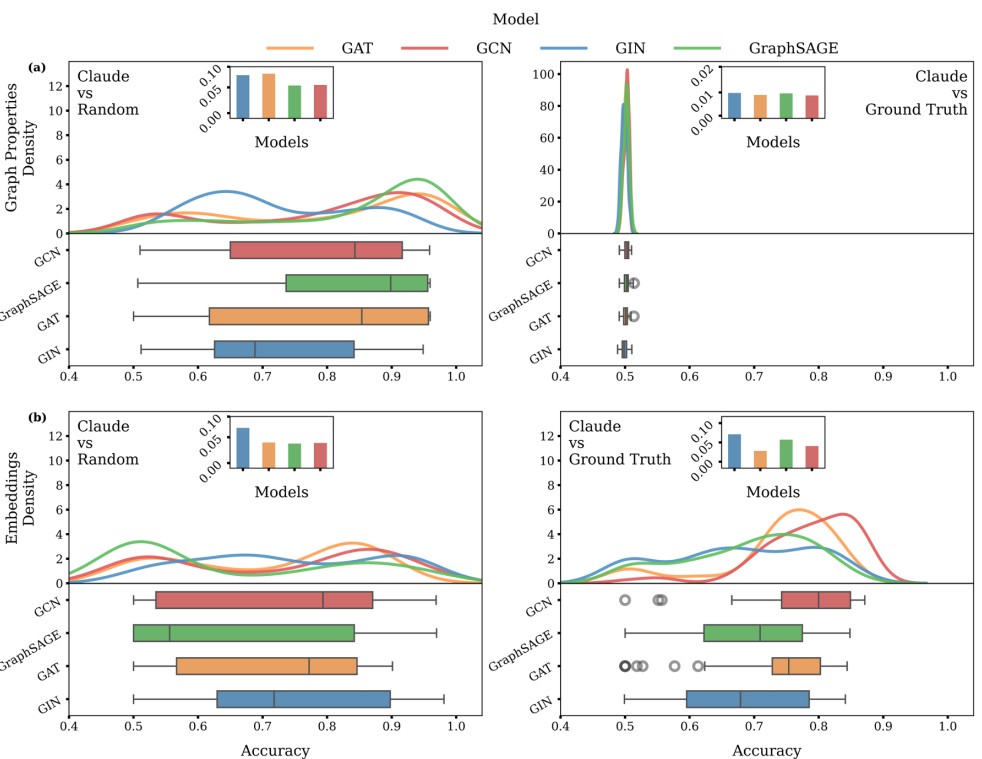

Figure 9: **Performance of the GNN for Claude dataset using SPECTER embeddings.** Each panel summarizing the performance of four GNN architectures on validation set across our binary classification tasks using different seeds. The figures are related to the GNN results using SPECTER embedding vectors. Within every panel, the top axis shows kernel-density estimates (KDEs) of per-run final accuracies, and the bottom axis shows each GNN architecture boxplots medians, interquartile ranges, whiskers, and outliers.

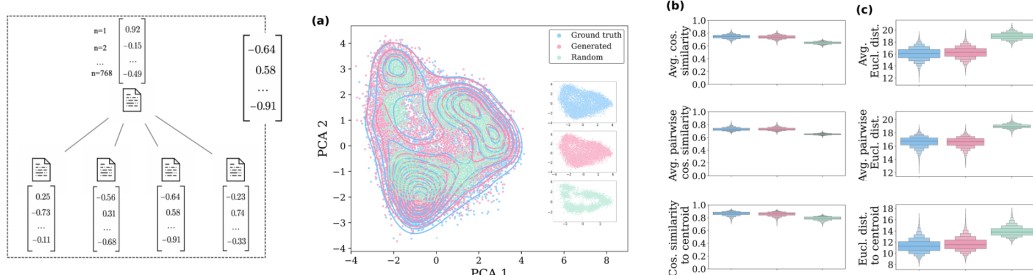

Figure 10: **PCA for GPT-4o graph embeddings with Cosine/EU Distances on SPECTER embeddings** Summing 768-d node embeddings to graph level, visualize the embedding space with 2D PCA (contours). The cosine alignment and Euclidean dispersion (node level) in three graph-wise alignment diagnostics: mean of focal with reference, mean of reference with reference, and focal vs. sum of references, capturing how well references align with the focal paper and with each other. Similar to graph properties, there is a slight change regarding the embedding features in the random subfield baseline.

| Title embeddings | Ground truth vs. GPT | Ground truth vs. Random | GPT vs. Random |
|---|---|---|---|
| Accuracy | $0.7502 \pm 0.0093$ | $0.8865 \pm 0.0060$ | $0.9344 \pm 0.0035$ |
| Mean F1-score | $0.7498 \pm 0.0093$ | $0.8854 \pm 0.0062$ | $0.9342 \pm 0.0035$ |

Table 8: **Performance of the RF on GPT-4o graphs using SPECTER.** The table shows the mean accuracy and F1-score obtained from applying a RF classifier across 10 independent runs with different random seeds, utilizing textual embeddings over graphs.

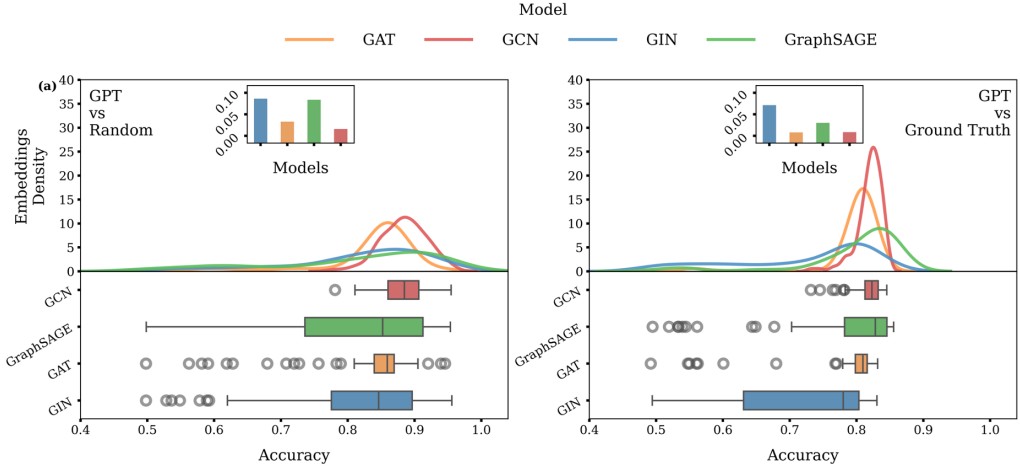

Figure 11: **Distribution of final validation accuracy over the sweep of hyperparameters using different models on GPT-4o graphs using SPCETER embeddings** Each panel summarizing the performance of four GNN architectures on validation set across our binary classification tasks using different seeds. The figures are related to the GNN results using embedding vectors. Within every panel, the top axis shows kernel-density estimates (KDEs) of per-run final accuracies, and the bottom axis shows each GNN architecture boxplots medians, interquartile ranges, whiskers, and outliers.

CROSS MODEL EXPERIMENT

We further probe cross-model generalization. Training GCN, GIN, GAT, and GraphSAGE on GPT-4o vs. ground-truth graphs (using OpenAI title embeddings) and then evaluating on test sets where GPT-4o graphs are replaced by Claude graphs yields accuracies around 0.68-0.80. These results suggest that the semantic fingerprint exposed by our models is to a large extent shared across GPT-4o and Claude, even though fine-grained citation patterns remain model-specific.

| Title embeddings | Ground truth vs. Generated |
|---|---|
| Mean Accuracy | $0.7205 \pm 0.0052$ |
| Mean F1-score | $0.7163 \pm 0.0055$ |

Table 9: **RF cross-model table.** Random Forest trained to distinguish ground-truth vs. generated citation graphs using summed 3072-dimensional title embeddings. Despite this generator swap, the classifier attains $\approx 0.72$ accuracy.

| Model | Accuracy | F1 |
|---|---|---|
| GCNNet | $0.8001_{\pm 0.0149}$ | $0.7947_{\pm 0.0166}$ |
| GINNet | $0.7619_{\pm 0.0194}$ | $0.7507_{\pm 0.0233}$ |
| GATNet | $0.6793_{\pm 0.0158}$ | $0.6500_{\pm 0.0231}$ |
| GraphSAGENet | $0.7042_{\pm 0.0125}$ | $0.6853_{\pm 0.0161}$ |

Table 10: **Results on test set for cross-model GNNs.** GCN, GIN, GAT, and GraphSAGE are trained to classify ground-truth vs. GPT-4o graphs using OpenAI title embeddings as node features, and then evaluated on a test set where the GPT-4o is replaced by Claude graphs. All architectures generalise substantially above chance , even though their detailed citation patterns are not seen during training.

RANDOM SUBFIELD BASELINE

As a robustness check, we provide the same analysis for classifying citation graphs using graph topology and LLM embeddings, but with a second, more fine-grained, random baseline. This Subfield-Random baseline restricts the sampling to papers within the same sub-field, provided that the sub-field contains at least 30 papers.

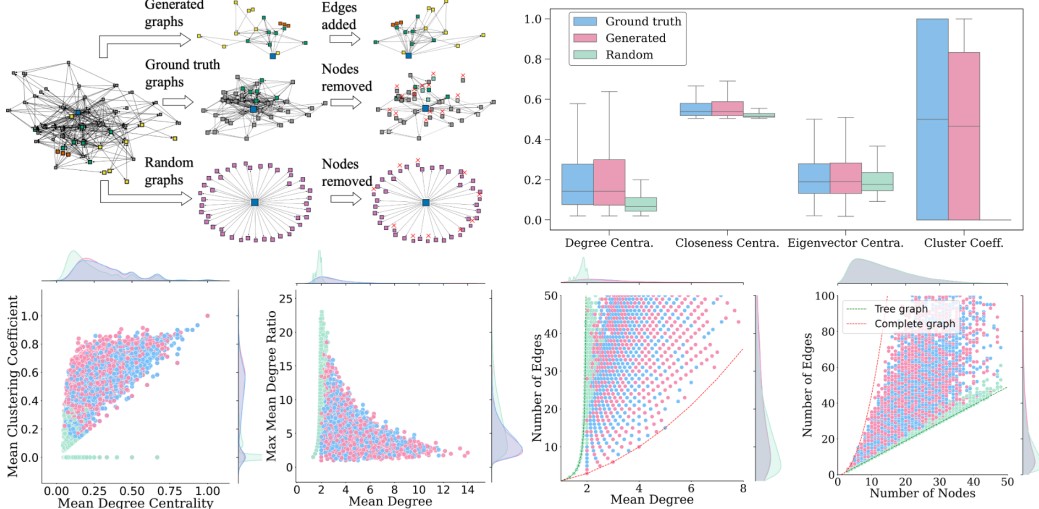

Figure 12: **Pipeline to generate the citation graphs and feature histograms** The box plot of distributions of four node-level metrics computed for each graph. Four scatter plots of joint features with marginal histograms where each point represents one graph, ground truth and generated graphs and random graphs. In all scatter plots ground truth and GPT-generated graphs overlap almost entirely whereas the random graphs exhibits differently, much like the random baseline in the same field that represent uniformly minimal connectivity, even though they were generated within the same sub-fields. The results exhibit quite similar properties to the random field shown in Figure 2.

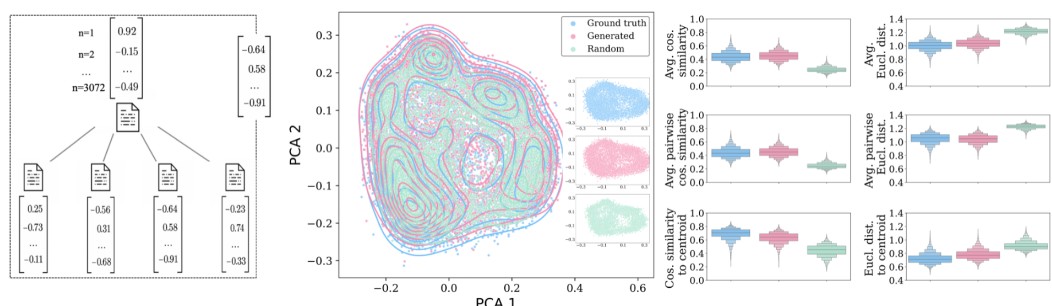

Figure 13: **PCA of graph embeddings with Cosine/EU Distances** Summing 3072-d node embeddings to graph level, visualize the embedding space with 2D PCA (contours). The cosine alignment and euclidean dispersion (node level) in three graph-wise alignment diagnostics: mean of focal with reference, mean of reference with reference, and focal vs. sum of references, capturing how well references align with the focal paper and with each other. Similar to graph properties, there is a slight change regarding the embedding features in the random subfield baseline.

## RANDOM TEMPORAL ORDERED PRESERVE

Here, we examine the temporal aspect of the random baseline in detail. In the original field-level reshuffling, temporal order (reference published before the focal paper) is in fact preserved for roughly 80% of focal–reference pairs, even though this was not enforced by construction. We built a new, stricter random baseline where we explicitly condition on publication year, so that focal papers are only assigned references from earlier years; under this construction, fewer than 1% of edges violate temporal order. For GPT-generated references themselves we find that about 6% of existing suggestions point to papers published after the focal paper. Importantly, using the temporally constrained baseline leaves our main conclusions unchanged: random graphs remain easy to separate from both ground truth and LLM graphs.

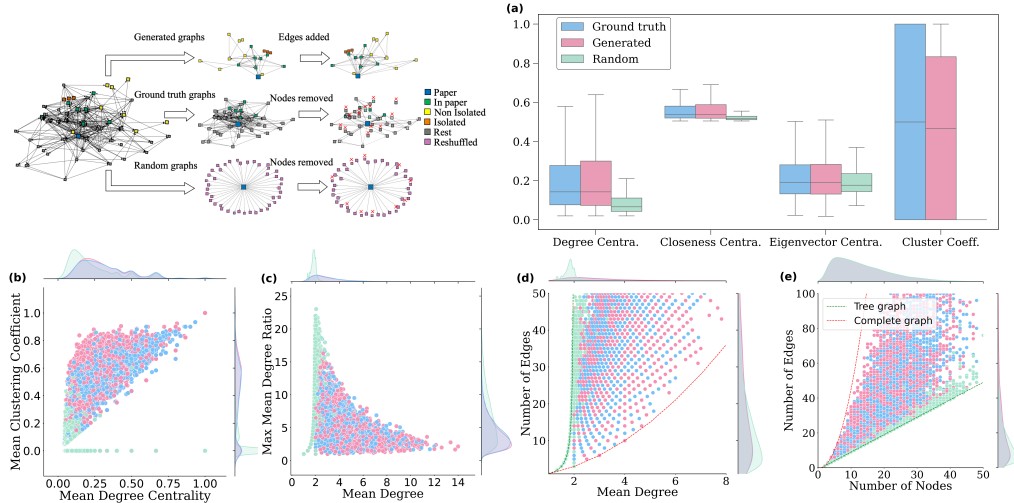

Figure 14: **structural comparison with temporally constrained random baseline.** Structural comparison between ground-truth citation graphs, GPT-generated graphs, and temporally constrained random graphs. In all scatter plots ground truth and GPT-generated graphs overlap almost entirely whereas the random graphs exhibits differently, much like the random baseline in the same field that represent uniformly minimal connectivity, even though they were generated within explicitly condition on publication year. Our main results remain unchanged, with random graphs still clearly separable from both ground truth and LLM-generated graphs.

| Graph properties | Ground truth vs. GPT | Ground truth vs. Random | GPT vs. Random |
|---|---|---|---|
| Mean Accuracy | $0.6119 \pm 0.0103$ | $0.8546 \pm 0.0064$ | $0.8837 \pm 0.0052$ |
| Mean F1-score | $0.6099 \pm 0.0101$ | $0.8533 \pm 0.0066$ | $0.8831 \pm 0.0051$ |

Table 11: **Performance of the RF using temporal preserved random baseline graph properties.** Random Forest performance on graph-level structural descriptors for distinguishing ground-truth, GPT-generated, and temporally constrained random citation graphs.

## RANDOM-VECTOR CONTROL AND PCA-k ABLATION

To test whether the accuracy gains are merely a side-effect of high dimensionality, we run two controls. First, we replace all semantic node embeddings with 3072-dimensional i.i.d. random vectors of matching dimensionality to GPT-generated references and ground-truth references : in this setting both RF and GNNs collapse to chance performance only by exploiting trivial size differences. Second, we project title embeddings to k principal components and retrain GNNs: train and validation accuracy closely tracks cumulative explained variance, dropping toward chance for small k and recovering only when we retain a large fraction of the semantic variance $d \in 8, 32, 512, 1024$. Together, these controls indicate that separability comes from semantic content rather than from the sheer number of embedding dimensions. For the RF, ground-truth–vs–generated accuracy drops to $0.5047$, i.e. at chance. This shows that what matters is not the raw dimensionality but the structured semantic information in the embeddings.

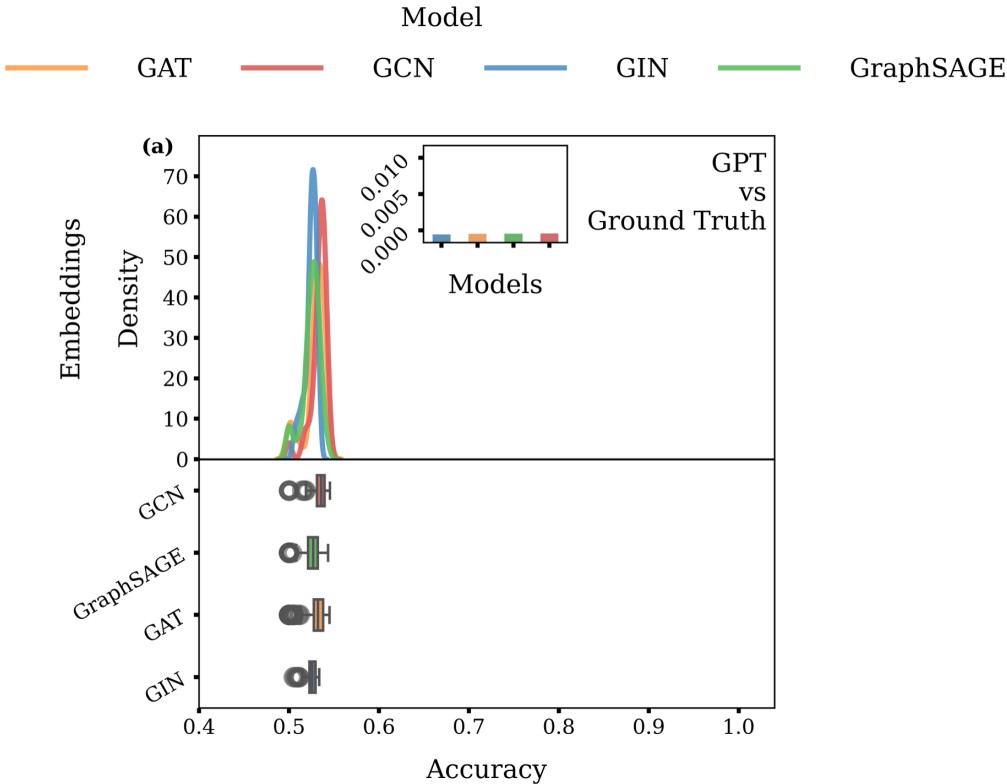

Figure 15: **Accuracy of GNN classifiers trained on node embeddings to distinguish GPT-generated from ground-truth citation graphs.** The inset reports the control experiment where semantic node embeddings are replaced by random vectors of the same dimensionality: in this setting, all architectures drop back to chance-level accuracy.

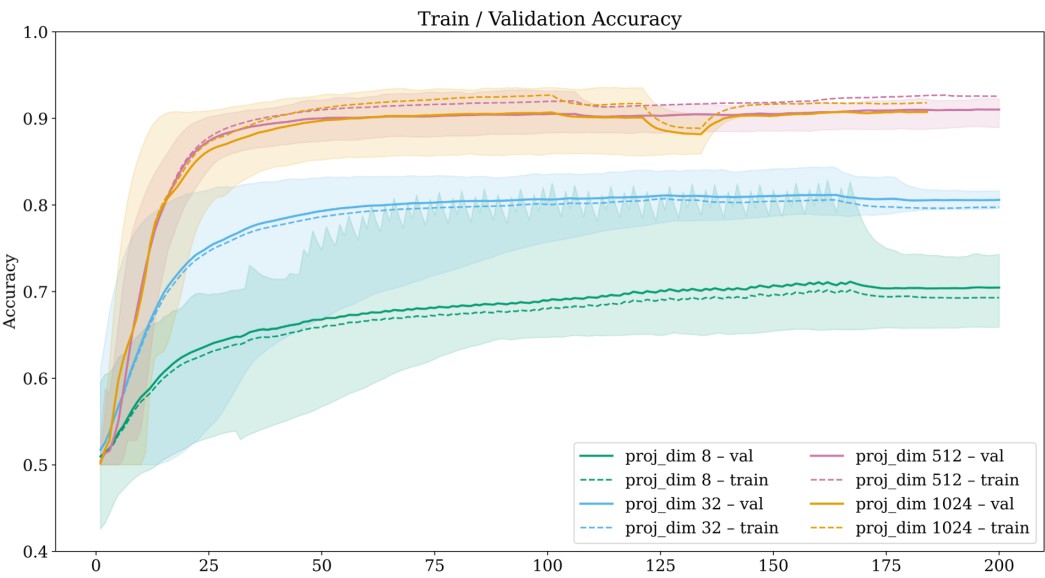

Figure 16: **Effect of embedding projection dimension on classification accuracy** Train and validation accuracy for GNN classifier distinguishing GPT-generated from ground-truth citation graphs when the 3072-dimensional title embeddings are linearly projected to smaller dimensions. Curves show the mean across seeds. This dimensionality-controlled ablation indicates that the advantage of semantic embeddings is not merely due to having more features, but to the information they encode about citation content.

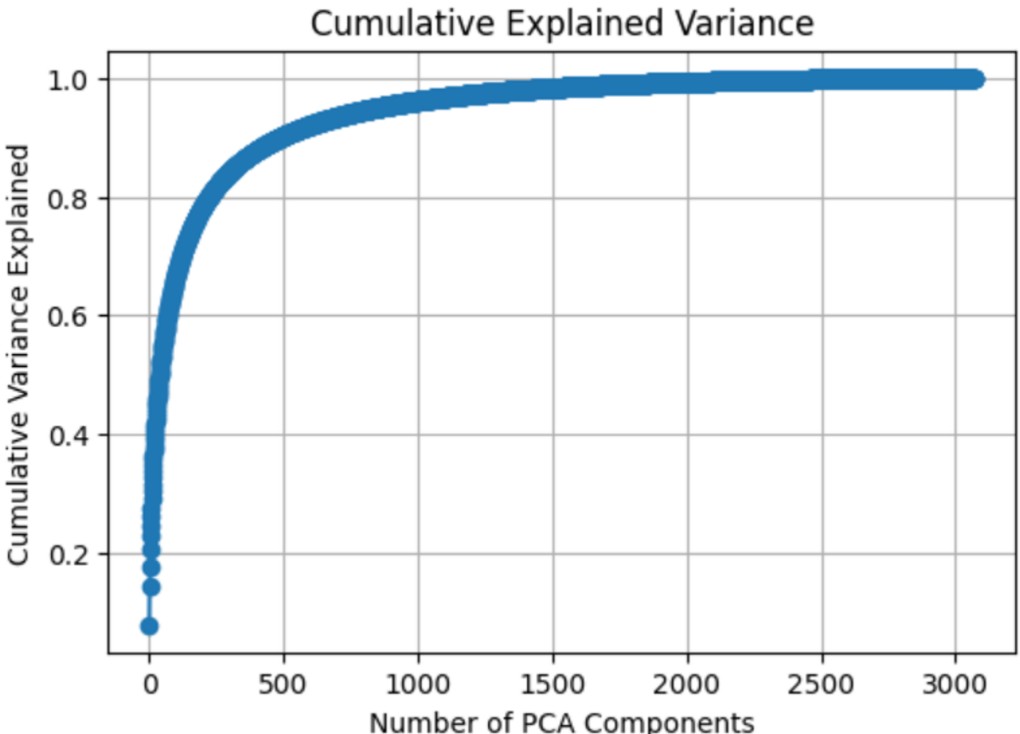

Figure 17: **Cumulative explained variance of title-embedding PCA.** Cumulative fraction of variance explained by principal components of the 3072-dimensional title embeddings, aggregated over all focal and reference papers. The first two components used in the 2-D visualization of Figure 3 capture only a small fraction of the total variance (6%), and the curve continues to rise steadily. A few hundred components explain over 90%. This confirms that the 2-D PCA plot is a heavily compressed view, while classifiers operate in a much higher-dimensional semantic subspace.

ISOLATED NODE SEMANTIC ROLE

We examine the semantic role of isolated GPT-generated references by measuring cosine similarities between their embeddings and the rest of the graph.

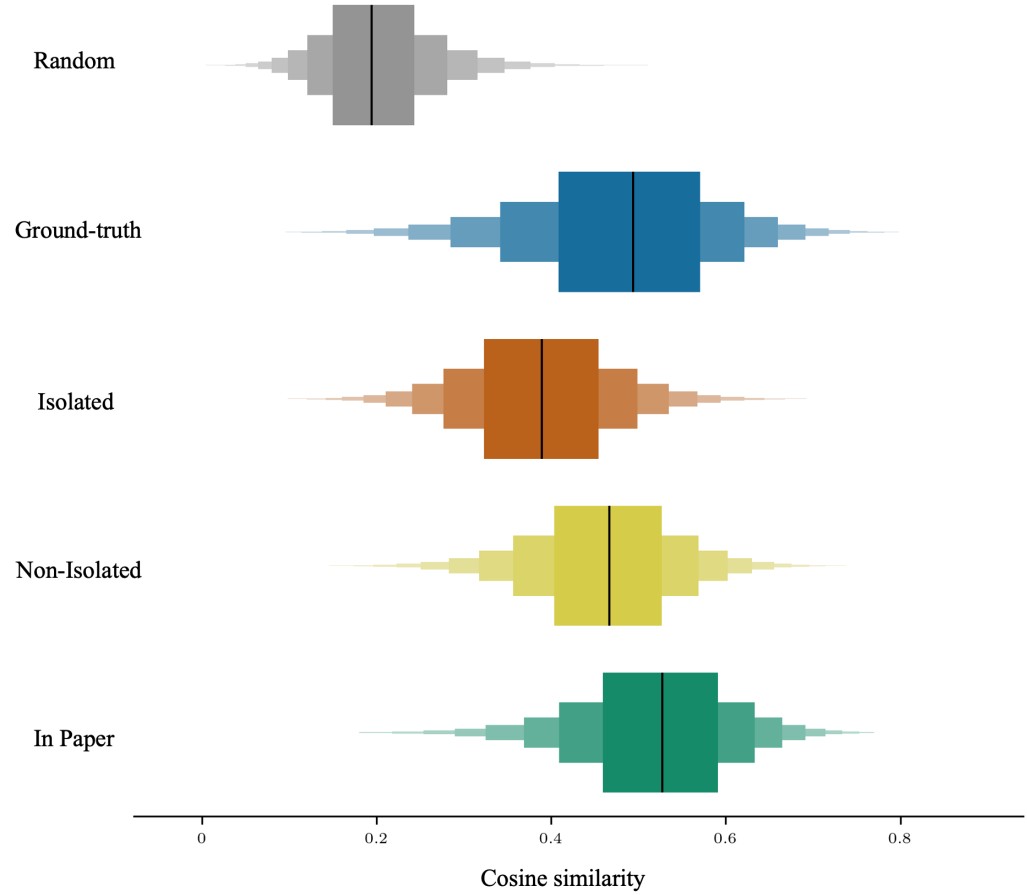

Figure 18: **Semantic role of isolated nodes** For each reference node, we compute the mean cosine similarity between its title embedding and the embeddings of all other nodes (including the focal paper) in the same graph. The box violin indicates the empirical distribution with the median marked in black. Random references have substantially lower similarity and a narrower spread, while all three GPT categories are much closer to ground truth. Isolated nodes sit slightly below non-isolated and in-paper references but remain far from the random baseline, suggesting that they are semantically plausible but more peripheral additions rather than off-topic hallucinations.

HYPERPARAMETER SETUP

Table 12: Hyperparameter search space and training setup for GNN experiments.

| Hyperparameter | Search Space / Setting |
|---|---|
| Batch Size | {8000, 10000, 13000} |
| Hidden Dimensions | {32, 64, 128} |
| Number of GNN Layers | {1, 2, 3, 4} |
| Learning Rate | Continuous in $[10^{-4}, 10^{-2}]$ |
| Dropout Rate | Continuous in $[0.0, 0.5]$ |
| Weight Decay | Continuous in $[0.0, 0.01]$ |
| Training Epochs | Maximum of 500 |
| Early Stopping | Patience = 15 epochs (no validation improvement) |
| Tuning Strategy | Random search per architecture/task |

Table 13: Final hyperparameters (Ground truth vs GPT)

| Hyperparameter | Graph properties | | | | Embeddings | | | |
|---|---|---|---|---|---|---|---|---|
| | GCN | GraphSage | GAT | GIN | GCN | GraphSage | GAT | GIN |
| Learning rate $(10^{-4})$ | 47.3668 | 8.8814 | 10.2697 | 47.3791 | 1.8859 | 5.7781 | 51.0714 | 15.4030 |
| Hidden dim | 64 | 32 | 64 | 32 | 128 | 128 | 128 | 64 |
| Dropout | 0.1494 | 0.2391 | 0.0003 | 0.0026 | 0.3582 | 0.4984 | 0.3426 | 0.2535 |
| Num layers | 1 | 1 | 1 | 1 | 4 | 3 | 4 | 3 |
| Weight decay | 0.0006 | < 0.0001 | 0.0003 | 0.0026 | 0.0054 | 0.0049 | < 0.0001 | 0.0005 |
| Batch size | 13000 | 8000 | 8000 | 13000 | 8000 | 10000 | 8000 | 8000 |

Table 14: Final hyperparameters (Random vs GPT)

| Hyperparameter | Graph properties | | | | Embeddings | | | |
|---|---|---|---|---|---|---|---|---|
| | GCN | GraphSage | GAT | GIN | GCN | GraphSage | GAT | GIN |
| Learning rate $(10^{-4})$ | 48.4246 | 19.2918 | 69.6247 | 26.3361 | 84.7239 | 94.0900 | 27.2811 | 15.4219 |
| Hidden dim | 32 | 64 | 64 | 32 | 32 | 32 | 32 | 128 |
| Dropout | 0.0312 | 0.2455 | 0.2268 | 0.0075 | 0.4938 | 0.2636 | 0.0621 | 0.4876 |
| Num layers | 4 | 1 | 1 | 1 | 3 | 2 | 3 | 2 |
| Weight decay | 0.0006 | 0.0002 | 0.0010 | 0.0013 | 0.0006 | 0.0002 | 0.0016 | 0.0001 |
| Batch size | 13000 | 8000 | 13000 | 10000 | 10000 | 13000 | 8000 | 8000 |

## Saturation analysis

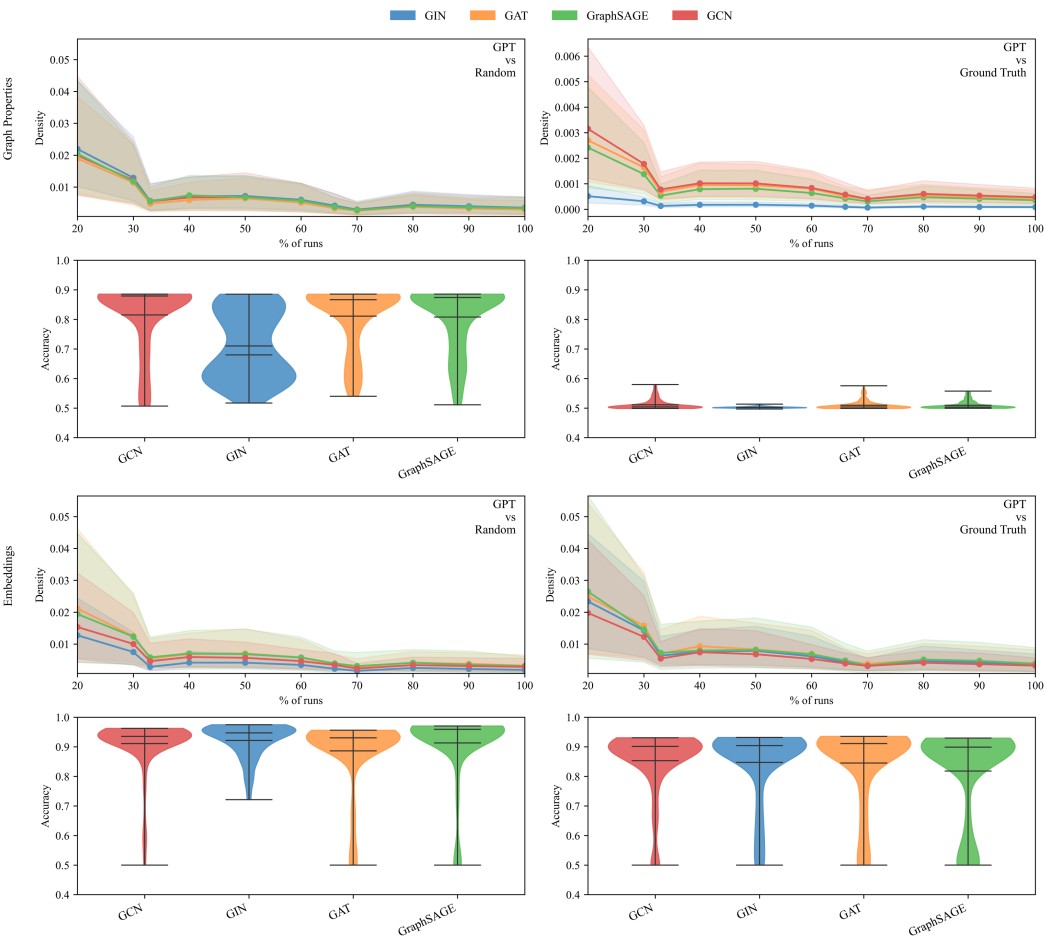

Figure 19: **Saturation analysis across different models** Panels are arranged by task (rows: Graph Properties, Embeddings) and supervision (columns: GPT vs Random, GPT vs Ground Truth). Top panels: permutation-averaged Wasserstein-1 distance between successive cumulative subsets of runs (20-100%). Curves are averaged over 500 random permutations; Smaller values and plateaus show distributional saturation. Across settings, the Wasserstein distance drops sharply by $\approx$ 30-40% of runs and flattens beyond $\approx$ 60-70%, suggesting the additional runs provide only a limited marginal benefit; model-wise convergence rates are broadly similar, with differences reflected in the accuracy distributions as shown in the bottom panels.

RANDOM FOREST

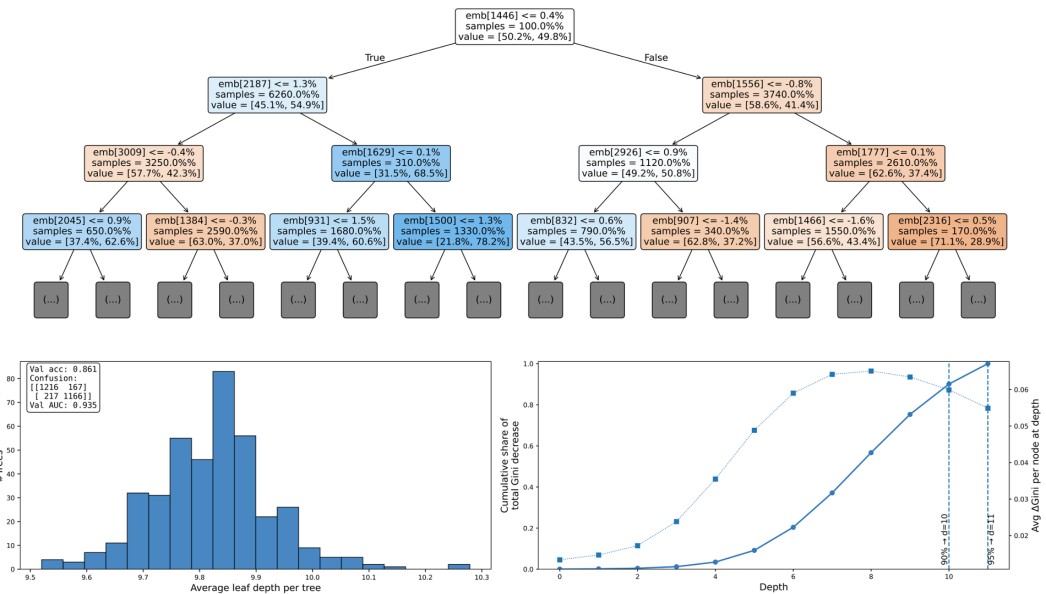

Figure 20: **Random-forest overview** Example decision tree illustrating the early splits on embedding features where Gini is node impurity. Boxes use separate colors per class where blue presenting GPT class and orange presenting the ground truth class. Histogram of the average leaf depth per tree in the forest showing that most trees are relatively shallow, nearly 9 levels on average; Cumulative share of total Gini decrease as a function of depth, indicating that the vast majority of impurity reduction occurs within the first $\approx$ 10–11 levels.

