# OpenReview forum: "Structurally Human, Semantically Biased: Detecting LLM-Generated References with Embeddings and GNNs"
_ICLR.cc/2026/Conference — ICLR 2026 Poster_

### Official Review · Reviewer_XBHR · 2025-11-01

**Soundness:** 2
**Presentation:** 3
**Contribution:** 3
**Rating:** 4
**Confidence:** 3

**Summary:**

The paper investigates whether citation graphs can distinguish between reference generated by LLM and those by humans. The author uses SciSciNet constructs 3 sets of paired graphs for each paper: the original graph, a graph generated by GPT-4o and a random graph. The research finds that only using structural features is difficult to distinguish GPT from humans, but the semantic information can increase the accuracy of distinguishing. Both human and GPT graphs are easily distinguished from random..

**Strengths:**

1. The story is interesting and timely, the work empirically investigates whether LLM-generated bibliographies hold detectable signals distinct from human ones.
2. The paper has clear experimental design. For each paper, the authors build three paired graphs (human / GPT / random) and control reference count, field, and year. This makes it easy to see what structure explains and how much semantics adds.

**Weaknesses:**

Although the limitations are discussed fairly comprehensively, several issues should be addressed to make the work more well-rounded:
1. Since the paper is about LLM capability, evaluating only GPT-4o is insufficient. Cross-model experiments are needed (e.g., generate with other LLMs; keep the embedding backbone fixed). Conversely, keep the generator fixed and vary the embedding backbone.
2. No detailed analysis about the LLM bias. A literature field analysis would reveal performance differences across different disciplines, and a temporal analysis could test for newer vs. older references.
3. Only a small subset of basic structural features is used in this paper. Will it omit distinguishable structural signals, and what motivate the selection of these features?
4. Missing metadata normalization policy. The paper does not specify how reference fields are standardized/canonicalized before matching and analysis

**Questions:**

See my weaknesses above. In addition, lines 132–137: why replace each directed edge with an undirected one? This removes information about “who came first” and “who cited whom,” which I think is the essential structural signal.

---

> ### Author Response · Authors · 2025-11-21
>
> Weaknesses:
> Although the limitations are discussed fairly comprehensively, several issues should be addressed to make the work more well-rounded:
>
> 1) Since the paper is about LLM capability, evaluating only GPT-4o is insufficient. Cross-model experiments are needed (e.g., generate with other LLMs; keep the embedding backbone fixed). Conversely, keep the generator fixed and vary the embedding backbone.
>
> We agree that evaluating only GPT-4o with its own embedding backbone is insufficient for broad claims about LLM capability. Following this comment we have extended the experiments along both axes. On the generator side we now include Claude in addition to GPT-4o, building matched ground-truth / LLM / random graphs for both models. On the embedding side we decouple generator and encoder: GPT-4o–generated references embedded with SPECTER yield RF accuracy 0.7789 ± 0.0029 (Ground truth vs Generated), while Claude-generated references embedded with the OpenAI model yield 0.7688 ± 0.0067. For Claude graphs using only structural features, RF accuracies are 0.5389 ± 0.0084 (Ground truth vs Generated) and ≈0.96–0.97 against the random baseline, mirroring the GPT pattern. We are currently running GNNs and will include a summary table in the final version over the full grid of (generator, embedder) = {Claude, GPT-4o}×{OpenAI, SPECTER}. All results so far support the main conclusion: structural realism is robust across models, while a semantic fingerprint remains.
>
>
> 2) No detailed analysis about the LLM bias. A literature field analysis would reveal performance differences across different disciplines, and a temporal analysis could test for newer vs. older references.
>
> We thank the reviewer for encouraging a deeper analysis of LLM biases. We will add a discussion on related works about the takeaways and the discussions.
> We will expand the temporal analysis described in our response to Reviewer 1 and summary statistics.
>
>
> 3) Only a small subset of basic structural features is used in this paper. Will it omit distinguishable structural signals, and what motivates the selection of these features?
>
> It is true that we only use a relatively small set of structural descriptors. This is a deliberate choice. The space of plausible structural features is extremely large, but our primary goal is to cleanly separate “structure” from “semantics” using interpretable, widely-used quantities that already capture the main aspects of citation topology considered in the Algaba et al. work. Within that category we believe we have included the relevant, non-trivial signals. The fact that even this reasonably rich set fails to distinguish GPT/Claude from ground truth, while semantic features succeed, strengthens the core claim that the key differences lie in content rather than in coarse graph structure.
>
>
> 4) Missing metadata normalization policy. The paper does not specify how reference fields are standardized/canonicalized before matching and analysis.
>
> In line with SciSciNet (Lin et al., 2023), field labels are derived from the MAG (Microsoft Academic Graph) fields-of-study records: we use the Level-0 fields as “top fields” and the Level-1 fields as “subfields”, based on the field assignments provided in the SciSciNet_PaperFields table. When a paper has multiple field entries, we keep the field with the highest normalized citation count, implicitly assuming that this field is more relevant to that paper; this is how we resolve ambiguity in field assignments. We will add a short subsection describing this field-disambiguation step.

---

> ### Author Response · Authors · 2025-11-21
>
> Questions:
> 1) Why replace each directed edge with an undirected one? This removes information about “who came first” and “who cited whom,” which I think is the essential structural signal.
>
> We appreciate this thoughtful point. Our decision to symmetrize citation edges, yielding a simple graph, was to analyze the local reference neighborhood of each focal paper to compare organizational topology, rather than arising from edge direction or trivial in-/out-degree differences.

---

> ### Author Response · Authors · 2025-12-04
>
> In line with our earlier reply, we have now added a set of broader cross-model, cross-encoder, and bias-related analyses. First, we extended the study to Claude as a second generator and evaluated detectors using multiple encoder configurations: structure-only RF, OpenAI-based embeddings, SPECTER-based embeddings, and GNNs over both embedding spaces. We also added a cross-encoder RF experiment where SPECTER embeddings are applied to GPT-4o graphs. Across these settings, ground-truth and LLM graphs remain hard to separate from each other using coarse topology but are cleanly separable from random baselines, while semantic embeddings consistently reveal a robust fingerprint—confirming that our conclusions do not depend on a specific generator–encoder pair. Second, to probe cross-model robustness, we trained RF and four GNN architectures on GPT-4o graphs (OpenAI embeddings) and evaluated them on Claude graphs, observing substantial but imperfect transfer well above chance. Third, we expanded the temporal-bias analysis by including year histograms and a temporally constrained random baseline with publication-year ordering preserved, plus corresponding RF results; these show that even with realistic time structure, random graphs remain clearly distinct from both ground truth and LLM graphs. Together, these experiments substantiate the broader-coverage and bias-oriented checks you asked for and reinforce the main claim that structural realism is robust across models and encoders, while a stable semantic fingerprint persists in the embeddings.

---

### Official Review · Reviewer_cRu8 · 2025-11-02

**Soundness:** 3
**Presentation:** 2
**Contribution:** 4
**Rating:** 6
**Confidence:** 4

**Summary:**

### Summary

This study presents an approach to assess whether reference lists generated by large language models (LLMs) can be distinguished from those created by humans. The authors construct paired citation graphs using 10,000 papers from SciSciNet.

To generate synthetic reference lists, they use GPT-4o, providing paper metadata such as title, abstract, authors, year, and venue. Three citation graphs are created:

- (A) Human ground-truth references (extracted from each paper)
- (B) GPT-generated references
- (C) Field-matched random baseline (references randomly sampled from papers in the same field and year)

---

### Methodology

Using these three citation graphs, the authors introduce three classification models to determine whether a reference list was generated by GPT-4o or by humans. These are applied as pairwise binary classification tasks (A vs B, A vs C, B vs C):

1. **Graph Topology:**
   A Random Forest (RF) classifier trained on hand-crafted graph-level statistics to distinguish citation networks by their structural topology.

2. **Semantic Embeddings:**
   An RF classifier using aggregated text-embedding vectors of paper titles to classify graphs based on the semantic coherence of their reference papers.

3. **Graph Neural Networks (GNNs):**
   Four GNN classifiers that jointly learn from citation structure and node embeddings through message passing, capturing both topological and semantic features.

---

### Key Findings

1. The models achieved an accuracy of 0.6 when distinguishing between (A) and (B), showing that GPT mimics the structural patterns of human references.
2. Adding semantic features improved discrimination between (A) and (B), suggesting that main differences lie in semantics.
3. Structure-only GNNs could not effectively distinguish GPT-generated from human references, while embedding-based GNNs performed well when distinguishing (B) from (C).

**Strengths:**

### Strengths

- **Originality:**
  This paper addresses the novel problem of distinguishing LLM-generated reference lists from human ones by constructing citation graphs and developing three classification approaches. While citation graphs and Random Forests are established methods, their application to this specific problem represents a new use case.

- **Quality:**
  The study demonstrates a solid experimental setup by evaluating three distinct classification approaches across multiple models. The authors leverage a large-scale dataset of 10,000 papers and 275,000 references, enabling comprehensive and meaningful analysis.

- **Clarity:**
  The paper is well-written and logically structured. Each approach is supported with clear explanations and informative figures.

- **Significance:**
  The paper tackles a timely research question with little prior work. While earlier studies have examined the detection of LLM-generated content in general, the detection of LLM-generated references remains largely unexplored. This study thus provides a valuable foundation for future research.

**Weaknesses:**

### Weaknesses

- **Related Work:**
  The paper lacks a dedicated related work section and does not fully contextualize its contribution. The authors briefly mention prior studies by Algaba et al. (2024), Mobini et al. (2025), and Algaba et al. (2025), but do not clarify the distinct contributions of each. A more thorough discussion is needed, especially regarding connections to related areas such as LLM-generated content detection, citation network analysis, citation generation, citation recommendation, and citation-related biases.

- **Discussion:**
  Although the results of the three classification approaches are presented clearly within individual sections, the paper lacks a consolidated discussion that synthesizes the findings and compares the approaches. A dedicated discussion section would enhance interpretability and depth.

**Overall:**
The paper could be significantly improved by adding comprehensive related work and discussion sections.

**Questions:**

**Major suggestions**
1. Include a comprehensive related work section.
2. Add a discussion section summarizing and comparing the key findings.

**Minor suggestions**
1. **Figure 2:** Simplify the figure to reduce textual complexity and improve readability; this could also free up space for expanded related work and discussion sections.
2. **Table 3:** Adjust the table to fit within the page margins specified by the template.
3. **References:** Add DOIs or URLs to allow readers to verify sources more easily.
4. **Grammar:** Correct the minor typo in line 465 (“have have”).

---

> ### Author Response · Authors · 2025-11-21
>
> Weaknesses:
>
> 1) The paper lacks a dedicated related work section and does not fully contextualize its contribution. The authors briefly mention prior studies by Algaba et al. (2024), Mobini et al. (2025), and Algaba et al. (2025), but do not clarify the distinct contributions of each. A more thorough discussion is needed, especially regarding connections to related areas such as LLM-generated content detection, citation network analysis, citation generation, citation recommendation, and citation-related biases.
>
> We thank the reviewer for this helpful suggestion. In the revision we will add a dedicated related-work section.
>
> 2) Although the results of the three classification approaches are presented clearly within individual sections, the paper lacks a consolidated discussion that synthesizes the findings and compares the approaches. A dedicated discussion section would enhance interpretability and depth.
>
> We will add a dedicated discussion section that synthesizes the results across the three modeling layers (structural RF, embedding-based RF, and GNNs) as well as the new cross-model experiments. This section will explicitly summarize the overarching pattern (random graphs are easily rejected; GPT/Claude graphs are structurally human-like but semantically biased). This should provide the higher-level synthesis that is currently missing.

---

> > ### Author Response · Authors · 2025-11-21
> >
> > Questions:
> >
> > 1) Simplify the figure to reduce textual complexity and improve readability; this could also free up space for expanded related work and discussion sections.
> >
> > We appreciate the suggestion. We will streamline Fig. 2 in the revised version.
> >
> > 2) Adjust the table to fit within the page margins specified by the template.
> >
> > We will adjust the table in the revised version.
> >
> > 3) Add DOIs or URLs to allow readers to verify sources more easily.
> >
> > We will add DOIs or stable URLs across the bibliography to improve verifiability.
> >
> > 4) Correct the minor typo in line 465 (“have have”).
> >
> > We will fix the typo in the revised version. Thanks for your precise attention to this matter.

---

### Official Review · Reviewer_13Z6 · 2025-11-02

**Soundness:** 2
**Presentation:** 3
**Contribution:** 2
**Rating:** 6
**Confidence:** 3

**Summary:**

The paper investigates whether large language models like GPT-4o produce bibliographies that can be distinguished from real, human-authored bibliographies when viewed through the lens of citation graph analysis.  The authors construct a large paired dataset of 9,218 citation graphs from SciSciNet, each with a ground-truth set of references, an equal-sized GPT-generated list, and a random baseline. The experiments show that using structural information alone cannot separate GPT-generated citations from the ground truth using either Random Forests or GNNs. However, when using semantic embeddings (from the title and sometimes the abstract), both RF and GNNs can distinguish the GPT-generated citations from the ground truth.

**Strengths:**

1. The findings of the paper are timely, as LLM-generated scientific bibliographies are emerging. Knowing the bias in the generated citations would be helpful to all researchers (and reviewers).
2. The paper provides firm empirical grounding. Not only does the paper present the accuracy results, but it also includes descriptive statistics and visualization to confirm/motivate the results (Figures 2 and 3).
3. The paper presents a carefully controlled dataset of ~9k citation graphs across three conditions. The results from two distinct machine learning methods (tree-based models and many GNN architectures) align closely.

**Weaknesses:**

1. The paper focuses on "parametric knowledge" and GPT-4o. This setting might not reflect all LLMs' behaviors nor citation recommendation systems that increasingly perform "deep research".
2. While the paper argues that semantic embeddings reveal the key differences between human and GPT-generated citation graphs, this claim is potentially confounded by the extreme disparity in feature dimensionality (3072 vs 5). The observed accuracy gain in the Random Forest might partly reflect model capacity rather than genuine informational superiority. A dimensionality-controlled ablation or matched-feature comparison would strengthen this conclusion.
3. There was some ambiguity in the text.
    - The baseline citations are constructed based on the field of the papers. However, it is unclear whether the constructions follow the citations' chronological order.
    - The paper states that the citation graphs of the three datasets have the same degree. However, summing over vectors is still an unusual option.
    - Figure 3 shows that the projected embedding of the GPT and ground truth citations overlap, but the classification model can distinguish between them. This means that 2D PCA lost most of the variance.  Thus, it might be helpful to see the plot.

**Questions:**

1. Was temporal order preserved in the random baseline graphs?
2. What percentage of variance is captured by the PCA projection in Figure 3? Perhaps t-SNE would be a better choice?
3. Is the embedding (semantic of structure) learnable under the GNN models?
4. If we append a random vector of size 3,067 to the structural node embedding, would it improve the performance of the GNNs?

---

> ### Author Response · Authors · 2025-11-21
>
> Weaknesses:
>
> 1) The paper focuses on "parametric knowledge" and GPT-4o. This setting might not reflect all LLMs' behaviors nor citation recommendation systems that increasingly perform "deep research".
>
> We appreciate the suggestion. We deliberately focus on the “parametric knowledge” regime, i.e. GPT-4o proposing references without access to external tools or databases, because this gives a clean lab setting. We agree that this does not cover retrieval-augmented “deep research” systems or all LLM families. The new Claude experiments partially broaden the generator side, but a systematic evaluation of tool-augmented citation recommenders is deliberately beyond the scope of this work. We will clarify this framing in the introduction and limitations, and explicitly point to retrieval-based setups as an important direction for future work.
>
> 2) While the paper argues that semantic embeddings reveal the key differences between human and GPT-generated citation graphs, this claim is potentially confounded by the extreme disparity in feature dimensionality (3072 vs 5). The observed accuracy gain in the Random Forest might partly reflect model capacity rather than genuine informational superiority. A dimensionality-controlled ablation or matched-feature comparison would strengthen this conclusion.
>
> To check whether the advantage of semantic embeddings is merely a dimensionality effect, we ran two extra ablation experiments. First, we applied PCA to the 3072-D embeddings and retrained the GNN for decreasing numbers of principal components. As we reduce k, accuracy falls in parallel with the explained variance: when only a small fraction of the variance is retained, performance is close to chance; high accuracy reappears only once a substantial portion of the variance is kept. Second, we assigned each node an i.i.d. random vector with the same dimensionality as the original embedding and reran the RF. In this case, Ground truth vs Generated accuracy drops to 0.5047, i.e. at chance. We will also provide the results of GNN using the random vector (currently running). This shows that what matters is not the raw dimensionality but the structured semantic information in the embeddings.

---

> ### Author Response · Authors · 2025-11-21
>
> Weaknesses:
>
> 3) There was some ambiguity in the text.
>
> a) The baseline citations are constructed based on the field of the papers. However, it is unclear whether the constructions follow the citations' chronological order.
>
> Following this question, we have examined the temporal aspect of the random baseline in detail. In the original field-level reshuffling, temporal order (reference published before the focal paper) is in fact preserved for roughly 80% of focal–reference pairs, even though this was not enforced by construction. To remove any ambiguity, we built  a new, stricter random baseline where we explicitly condition on publication year, so that focal papers are only assigned references from earlier years; under this construction, fewer than 1% of edges violate temporal order. For GPT-generated references themselves we find that about 6% of existing suggestions point to papers published after the focal paper. Importantly, using the temporally constrained baseline leaves our main conclusions unchanged: random graphs remain easy to separate from both ground truth and LLM graphs as you see in the link below in which we plot the same figure as in the paper, now using the new Random baline(https://filebin.net/7ywflhp0cqj7x46n). We will add a clarifying sentence and report the temporal-baseline results in an appendix table.
>
> b) The paper states that the citation graphs of the three datasets have the same degree. However, summing over vectors is still an unusual option.
>
> We enforce equal degree across the three graph types because otherwise degree would become a trivial discriminative feature: if GPT-generated lists systematically had more or fewer references, classifiers could rely almost entirely on degree statistics. By fixing degrees we force the models to use more informative structural or semantic cues. As for summing vectors, this appears in two places. For the cosine-similarity diagnostics (average cosine similarity, average pairwise cosine similarity, cosine similarity to centroid) we report both “average embeddings then cosine” and “cosines then average” to avoid privileging a single aggregation rule. For the RF we need a fixed-size graph-level input, and summing node embeddings is a standard permutation-invariant readout, analogous to sum/mean pooling in GNNs when mapping node embeddings to a graph embedding. We will clarify both points in the methods section.
>
> c) Figure 3 shows that the projected embedding of the GPT and ground truth citations overlap, but the classification model can distinguish between them. This means that 2D PCA lost most of the variance. Thus, it might be helpful to see the plot.
>
> Yes, we agree. The apparent overlap of classes in the 2D PCA plot is a consequence of the very low variance captured by the first two components: together they explain at most ≈6% of the total variance. High separability in the full 3072-D space is therefore fully compatible with strong overlap in that plane. To address the dimensionality issue, we have rerun the GNN experiments on embeddings projected to k PCA components, varying k. This explicitly controls for feature dimensionality and tells us how many directions the model actually needs. As k decreases, accuracy drops together with the cumulative explained variance. Below, we added a figure showing explained variance vs PCA dimension, together with the corresponding accuracies, to make this point explicit.
> Following the reviewer’s suggestion, we will add a plot of cumulative explained variance vs PCA dimension and discuss the dimensionality-controlled ablations, making explicit that Figure 3 is intended only as a coarse visualization rather than an indicator that embeddings are not separable. (https://filebin.net/7ywflhp0cqj7x46n)

---

> ### Author Response · Authors · 2025-11-21
>
> Questions:
>
> 1) Was temporal order preserved in the random baseline graphs?
>
> We have examined the temporal aspect of the random baseline in detail. In the original field-level reshuffling, temporal order (reference published before the focal paper) is in fact preserved for roughly 80% of focal–reference pairs, even though this is not enforced by construction. To remove any ambiguity, we built an additional, stricter random baseline where we explicitly condition on publication year, so that focal papers are only assigned references from earlier years; under this construction, fewer than 1% of edges violate temporal order. For GPT-generated references themselves we find that about 6% of existing suggestions point to papers published after the focal paper. Importantly, using the temporally constrained baseline leaves our main conclusions unchanged: random graphs remain easy to separate from both ground truth and LLM graphs. We will add a clarifying sentence and report the temporal-baseline results in an appendix table.
>
> 2) What percentage of variance is captured by the PCA projection in Figure 3? Perhaps t-SNE would be a better choice?
>
> The first two PCA components used in Figure 3 capture at most ≈6% of the total variance. We will report this number directly in the caption and add a panel showing cumulative explained variance as a function of the number of components. This will make clear why a 2D projection alone cannot be used to evaluate separability, and why we rely on RF/GNN performance rather than on low-dimensional visuals.
>
> 3) Is the embedding (semantic of structure) learnable under the GNN models?
>
> Our current evidence suggests that GNNs are not simply memorising individual embeddings. Training and validation curves are well aligned, with no signs of overfitting. To further stress-test this, we are adding a control where we replace semantic node embeddings by random vectors of the same dimensionality and retrain the GNNs; in analogy with the RF ablation, we saw accuracies to drop to chance (0.50) when the models are indeed exploiting semantic structure rather than rote memorisation. We will report the results of GNNs in the revised version.
>
>
>
> 4) If we append a random vector of size 3,067 to the structural node embedding, would it improve the performance of the GNNs?
>
> Conceptually, this is very close to the random-embedding controls above: because the added dimensions are pure noise, a well-regularised GNN should learn to ignore them and fall back on the structural signal, yielding performance comparable to the structure-only case. As mentioned in the previous comment, we are adding a control where we replace semantic node embeddings by random vectors of the same dimensionality and retrain the GNNs, we will mention this thought experiment in the paper and include the corresponding GNN ablation.

---

> ### Author Response · Authors · 2025-12-04
>
> Following our previous response, we have now completed the additional controls aimed at disentangling parametric-knowledge effects, embedding dimensionality, and temporal structure in the baselines. First, to broaden beyond GPT-4o, we reproduced the analysis with Claude as generator and evaluated both structure-only and embedding-based detectors using OpenAI and SPECTER encoders. As anticipated, structural descriptors remain near chance for ground truth vs. Claude, while semantic embeddings yield robust separability of ground truth vs. LLM and both vs. random baselines. Second, to address the dimensionality confound, we added (i) a random-vector control, forcing RF and GNN performance back to chance, and (ii) a PCA-dimension ablation showing that GNN accuracy tracks cumulative explained variance and only recovers as more semantic variance is retained. Third, we expanded the temporal analyses by (a) plotting year histograms and summary statistics for ground-truth vs. GPT-generated references, and (b) introducing a temporally constrained random baseline where all random references satisfy “reference year ≤ focal year”; structurally and in RF performance this baseline remains clearly separable from both ground truth and LLM graphs, leaving our conclusions unchanged. We additionally report RF and GNN results when using SPECTER embeddings on GPT-4o graphs, further reinforcing that the observed effects stem from semantic content rather than a specific encoder architecture. Collectively, these additions implement the sanity checks you requested and support our claim that the detection signal is semantic, not merely high-dimensional or driven by temporal artefacts.

---

### Official Review · Reviewer_owXj · 2025-11-08

**Soundness:** 2
**Presentation:** 3
**Contribution:** 2
**Rating:** 4
**Confidence:** 4

**Summary:**

This paper investigates whether LLMs when tasked with generating bibliographic references for scientific papers using only parametric knowledge, reproduce human-like citation patterns. Using 10,000 focal papers and around 275,000 references from SciSciNet, the authors construct paired citation graphs: ground-truth reference networks, GPT-generated reference networks, and a field-matched random baseline. Then they evaluate structural graph properties (centrality, clustering, density, etc. ), semantic embedding signals (OpenAI text-embedding-3-large), and multiple Graph Neural Networks (GCN, GAT, GIN, GraphSAGE). Their key results demonstrate that structural topology alone cannot reliably distinguish GPT-generated from human citation graphs, semantic embeddings do reliably distinguish them.
Overall it is a well written paper.

**Strengths:**

The following are 3 key strong points of this paper:
1. Authors carried out rigorous experimental design by using paired graphs per focal paper and apply field-matched randomization to break subtle citation structure, making conclusions robust. This is one of the strongest points of this paper.

2. Authors have done a good job of clearly decomposing Structural vs. Semantic signals. This leads to better interpretability and diagnostic insights.

3. Their experiments provide strong empirical evidence at scale by considering 10,000 focal papers and 9,218 valid paired graphs; this is large for bibliometric graph learning and this adds confidence.

**Weaknesses:**

1. The analysis carried out in this paper is limited to only GPT-4o. Concept drift/divergence between models is noted but not tested. This is a serious limitation of this paper.

2. While semantic embeddings separate classes, the authors do not identify what semantic dimensions differ (recency, methodology, prestige, jargon)?

3. For GNNs, only titles and simple metrics are used. Full-text content or citation contexts might yield deeper insights.. Is there any reason for this?

**Questions:**

Below are a few comments to the authors:

(a) Please answer the above 3 weak points.

(b) The conclusion states differences lie in semantics, but the dimensions are not unpacked. Attention weights, probing tasks, or PCA factors would help interpretability. What are your thoughts on this? How do you go about addressing this issue?

(c) Generate references with multiple models (Claude, LLaMA, DeepSeek), then train detectors on one and test on others. How the results would look in this case?

(d) Considering temporal bias metrics would be nice. In other words, LLMs are known to over-cite recent literature. Is this really happening in your experiments? If yes, please quantify this directly.

(d) Authors mention isolated nodes but does not explore their semantic role.. It would be nice to understand what such isolated nodes mean?

---

> ### Author Response · Authors · 2025-11-21
>
> Weaknesses:
>
> 1) The analysis carried out in this paper is limited to only GPT-4o. Concept drift/divergence between models is noted but not tested. This is a serious limitation of this paper.
>
> We thank the reviewer for highlighting this important limitation. Following this concern, we have rerun the full pipeline with Claude as an additional generator, constructing ground-truth / Claude / random graphs in exactly the same way as for GPT-4o. On Claude graphs with structure-only features, a Random Forest (RF) reaches 0.5389 ± 0.0084 accuracy for Ground truth vs Generated, and cleanly separates both from the random baseline (Random vs Generated: 0.9701 ± 0.0024; Random vs Ground truth: 0.9601 ± 0.0039). We also decouple generation and embedding: for GPT-4o–generated graphs embedded with SPECTER we obtain 0.7789 ± 0.0029 accuracy (Ground truth vs Generated), and for Claude-generated graphs embedded with the OpenAI model we obtain 0.7688 ± 0.0067. This shows that the semantic fingerprint is not an artefact of a single generator/encoder pair. In the revised version we will include the same type of figures (https://filebin.net/7ywflhp0cqj7x46n) as in the current paper, but now replicated for the Claude and cross-encoder settings, so that readers can directly compare them with the original GPT-4o–only results. We are currently running the corresponding GNN experiments and will add a full generator×embedder table (GPT / Claude × OpenAI / SPECTER) in the final version; all results so far support the same picture as in the paper: LLM-generated graphs are structurally realistic but remain semantically distinguishable from ground truth.
>
>
> 2) While semantic embeddings separate classes, the authors do not identify what semantic dimensions differ (recency, methodology, prestige, jargon)?
>
> We fully agree that it would be very interesting to unpack which semantic dimensions drive separability (recency, venue prestige, methodology vs theory, jargon, etc.). However, off-the-shelf embeddings do not provide a one-to-one mapping between input attributes and coordinates, and systematically back-tracing individual dimensions would require (if possible) an additional layer of explainability tools (probing tasks, diagnostic classifiers) on top of the current pipeline. In the present paper we therefore focus on the more modest claim that semantic node embeddings as a whole contain enough information to reliably distinguish LLM-generated from human graphs. We do complement this with diagnostics on some obvious dimensions, such as publication year, which suggest that recency alone does not account for the observed separability. A full decomposition of the embedding space into interpretable factors is an orthogonal line of work, and we will point to this explicitly as a direction for follow-up research.
>
> 3) For GNNs, only titles and simple metrics are used. Full-text content or citation contexts might yield deeper insights.. Is there any reason for this?
>
> The choice to work with titles (and abstracts for focal papers) rather than full texts or citation contexts is mainly driven by data availability. For the vast majority of references in SciSciNet we do not have full texts at scale, whereas titles are consistently available. At the same time, this restriction makes the differentiation task strictly harder: we already obtain strong RF and GNN performance using only shallow title-level semantics, showing that the GPT vs ground-truth signal is present even at this coarse level. For human readers, having access to full texts or citation contexts would likely make it easier to spot LLM-generated references; the results we report should thus be viewed as a lower bound on what richer text representations could achieve.

---

> ### Author Response · Authors · 2025-11-21
>
> Questions:
> Below are a few comments to the authors:
>
> (a) Please answer the above 3 weak points.
>
> (b) The conclusion states differences lie in semantics, but the dimensions are not unpacked. Attention weights, probing tasks, or PCA factors would help interpretability. What are your thoughts on this? How do you go about addressing this issue?
>
> To address the dimensionality issue, we have rerun the GNN experiments on embeddings projected to k PCA components, varying k. This explicitly controls for feature dimensionality and tells us how many directions the model actually needs. As k decreases, accuracy drops together with the cumulative explained variance: if we keep only a few components, which together explain at most ≈6% of the variance, the GNNs operate close to chance; performance stabilizes only once a much larger number of components is retained. This also explains why the 2D PCA in Figure 3 shows strong overlap: the first two components simply do not capture the directions that drive separability. In the revised version we will include a plot of cumulative explained variance vs PCA dimension, together with the corresponding accuracies, to make this point explicit. More sophisticated interpretability tools such as attention-based attribution or probing classifiers could be layered on top of our pipeline, but implementing and validating them would require a substantial additional effort, beyond the scope of this work. We will add a figure showing explained variance vs PCA dimension, together with the corresponding accuracies, to make this point explicit.(https://filebin.net/7ywflhp0cqj7x46n)
>
> (c) Generate references with multiple models (Claude, LLaMA, DeepSeek), then train detectors on one and test on others. How the results would look in this case?
>
> We thank the reviewer for this suggestion. Building on the new Claude experiments mentioned above, we are extending the setup to the genuine cross-model transfer case: training RFs and GNNs on graphs generated by one LLM (e.g. GPT-4o) and evaluating on graphs generated by another (e.g. Claude), while keeping the embedding backbone fixed, and vice versa. Given that both GPT-4o and Claude graphs show the same qualitative pattern (structurally human-like but semantically separable from ground truth) we expect substantial cross-model generalization. We will report the full train-on-A / test-on-B matrix and a summary table of RF and GNN results over all (generator, embedder) combinations in the revised version.
>
>
> (d) Considering temporal bias metrics would be nice. In other words, LLMs are known to over-cite recent literature. Is this really happening in your experiments? If yes, please quantify this directly.
>
> Our dataset is constructed based on the dataset introduced in Algaba et al. (2025), and therefore inherits the same temporal distribution of ground-truth references. In that work, the authors show that, among LLM-generated references, existing (non-hallucinated) ones tend to be slightly older than the ground-truth references (the median year of ground truth and existing generated references are identical) , while hallucinated (non-existing) references skew somewhat more recent; the mixture of the two reproduces the temporal distribution of the ground truth closely, and the overall effect is small. In our experiments we therefore observe the same qualitative behaviour, with the caveat that we focus only on the existing generated references. We will clarify this point in the manuscript and explicitly connect the temporal pattern observed here to the previously reported bias.
>
>
> (e) Authors mention isolated nodes but does not explore their semantic role.. It would be nice to understand what such isolated nodes mean?
>
> We thank the reviewer for drawing attention to this interesting subset of nodes.In the revision we will analyse their semantic role by computing cosine similarities between isolated nodes and the remaining nodes in the same graph. This might allow us to distinguish between isolated nodes that are still semantically close to the focal paper (plausible but uncited suggestions) and those that are both structurally and semantically out-of-place. We will include these diagnostics.

---

> ### Author Response · Authors · 2025-12-04
>
> As announced in our earlier response, we have now run the additional simulations you requested. These new results address four points. First, to test dependence on GPT-4o and model drift, we replicated the full pipeline with Claude as generator, both at the structural level and in embedding space, and report RF and GNN performance for Claude with OpenAI and SPECTER encoders. Second, to clarify the role of the embeddings and the 2-D PCA visualization, we added (i) a random-embedding control, where replacing all node embeddings by i.i.d. noise drives RF and GNN accuracies back to chance, and (ii) a PCA-k ablation with the cumulative explained-variance curve and GNN train/validation trajectories for different projection dimensions. Third, to evaluate cross-model transfer, we trained RF and GNNs on GPT-4o graphs and tested on Claude graphs with a fixed OpenAI embedder, obtaining substantially above-chance generalization across generators. Finally, we quantified the semantic role of isolated nodes by comparing cosine-similarity distributions for random references, ground-truth references, non-isolated GPT references, isolated GPT references, and references shared between GPT and the ground-truth bibliography. Together, these experiments support the points raised in our previous reply: the semantic fingerprint is not tied to GPT-4o, depends on meaningful embedding information rather than dimensionality alone, generalizes across generators, and isolated nodes behave as plausible but more peripheral references rather than off-topic hallucinations.

---

### Author Response · Authors · 2025-12-04

Dear AC and reviewers,
This paper provides a rigorous, paired evaluation of LLM-written bibliographies in which we disentangle topology from content by building ground-truth, LLM, and field/temporally matched random reference graphs around the same focal papers. We show that structure-only detectors easily reject random graphs yet are near chance for LLM vs. human (≈0.61), while semantic embeddings make LLM vs. human reliably separable (≈0.83). Because bibliographies are the primary entry point of LLMs into scholarly drafting and review, an auditable test that pinpoints a semantic—rather than structural—deviation is both relevant and urgent. During the rebuttal period we executed the full set of analyses requested by the reviewers. The results from these additional analysis further confirm our original message and add a robustness to our results. For the convenience of the AC and reviewers we re-iterate the three major experiments we added.

First, we replicated the pipeline with Claude, decoupled generator and encoder (GPT-4o↔SPECTER; Claude↔OpenAI), and demonstrated substantial cross-model transfer when training on GPT-4o and evaluating on Claude across four GNNs. Secondly, we established the causal role of node embeddings by demonstrating graphs with that random embeddings are not distinguishable and by including a PCA-k ablation which shows that accuracy tracks the increase in explained variance. Thirdly, we added a temporally constrained random baseline (reference-year ≤ focal-year) and year-distribution summaries, which leave conclusions unchanged. To conclude we also analyzed isolated LLM nodes via cosine-similarity diagnostics (typically plausible but peripheral), and made clarified our experimental setup The revised PDF integrating all figures and tables is uploaded.

Taken together, our results suggest a simple operating principle: today’s LLMs can convincingly mimic the shape of citation, but not yet its semantics and it is therefore in this semantic fingerprint that reliable detection (and eventual correction) must operate.

We appreciate the efforts from the reviewers. Their requests for clarification and additional experiments have allowed us to robustly confirm our results within a broader setting. We are conscious that many submissions are in a similar position this round, and that arriving at a fair outcome across such volume requires substantial effort from the ACs which we sincerely appreciate.

The Authors

---

### Meta-Review · Area_Chair_qZeR · 2026-01-07

**Summary:**

This paper looks at the very relevant problem of detecting LLM generated references versus real references. To do this the paper builds paired citation graphs and trains models to distinguish between the two. The paper reports encouraging results on being able to tell generated references from real references.

The main concerns of the reviewers were as follows

1. The limitation to only one model GPT 4o. Multiple reviewers (owXj, 13Z6, XBHR)  expressed concerns about only considering references generated by one model.

2. The use and interpretation of the semantic embeddings. Specifically, reviewer owXj asks for which semantic directions help distinguish between the fake and real citations. Reviewer 13Z6 worries about the extreme feature disparity might be driving the improved performance.

**Reviewer Concerns:**

I believe the second concern about the semantic embeddings was thoroughly addressed. In particular, the rebuttal presented two new experiments to show that the higher model capacity was not the driving factor but the semantic structure in the embeddings. I believe this claim to be fully addressed.

I consider the first concern to be only partially addressed. The addition of Claude is good. Additionally, the addition of the cross model case is also strong. That is training on GPT 4o and testing with Claude. However, I only consider this partially addressed, as it only adds a second LLM. I would consider it fully addressed if a third LLM had been added.

**Reviewer Scores:**

I believe that reviewers cRu8 and 13Z6, who are currently voting to accept the paper are likely to continue to do so. Reviewers owXj concerns have been mostly addressed and I believe they are likely to raise their score to a 6. Lastly Reviewer XBHR could potentially still have concerns about LLM bias.

As such I believe that this paper is should be accepted.

---

### Decision · Program_Chairs · 2026-01-26

Accept (Poster)